# Low-level repressive histone marks fine-tune gene transcription in neural stem cells

**Arjun Rajan**[1†], **Lucas Anhezini**[1†‡], **Noemi Rives-Quinto**[1], **Jay Y Chhabra**[1], **Megan C Neville**[2], **Elizabeth D Larson**[3], **Stephen F Goodwin**[2], **Melissa M Harrison**[3], **Cheng-Yu Lee**[1,4,5,6]*

[1]Life Sciences Institute, University of Michigan-Ann Arbor, Ann Arbor, United States; [2]Centre for Neural Circuits and Behaviour, University of Oxford, Oxford, United Kingdom; [3]Department of Biomolecular Chemistry, University of Wisconsin-Madison, Madison, United States; [4]Department of Cell and Developmental Biology, University of Michigan Medical School, Ann Arbor, United States; [5]Division of Genetic Medicine, Department of Internal Medicine, University of Michigan Medical School, Ann Arbor, United States; [6]Rogel Cancer Center, University of Michigan Medical School, Ann Arbor, United States

**\*For correspondence:**
leecheng@umich.edu

[†]These authors contributed equally to this work

**Present address:** [‡]Institute of Biological Sciences and Health, Federal University of Alagoas, Maceió, Brazil

**Competing interest:** The authors declare that no competing interests exist.

**Abstract** Coordinated regulation of gene activity by transcriptional and translational mechanisms poise stem cells for a timely cell-state transition during differentiation. Although important for all stemness-to-differentiation transitions, mechanistic understanding of the fine-tuning of gene transcription is lacking due to the compensatory effect of translational control. We used intermediate neural progenitor (INP) identity commitment to define the mechanisms that fine-tune stemness gene transcription in fly neural stem cells (neuroblasts). We demonstrate that the transcription factor Fruitless[C] (Fru[C]) binds cis-regulatory elements of most genes uniquely transcribed in neuroblasts. Loss of fru[C] function alone has no effect on INP commitment but drives INP dedifferentiation when translational control is reduced. Fru[C] negatively regulates gene expression by promoting low-level enrichment of the repressive histone mark H3K27me3 in gene cis-regulatory regions. Identical to fru[C] loss-of-function, reducing Polycomb Repressive Complex 2 activity increases stemness gene activity. We propose low-level H3K27me3 enrichment fine-tunes gene transcription in stem cells, a mechanism likely conserved from flies to humans.

## Editor's evaluation

This is an important study that defines the role of the FruC transcription factor in key developmental decisions during neurogenesis in *Drosophila*. The authors combine genetics and genomic profiling to provide convincing evidence that FruC-regulated gene expression is correlated with changes in repressive histone marks. This study will be of wide general interest to the developmental biology field.

## Introduction

Expression of genes that promote stemness or differentiation must be properly controlled in stem cells to allow their progeny to transition through various intermediate stages of cell fate specification in a timely fashion (*Pollen et al., 2015*; *Bhaduri et al., 2021*; *Michki et al., 2021*; *Ruan et al., 2021*; *Dillon et al., 2022*). Exceedingly high levels of stemness gene transcripts that promote an undifferentiated

**eLife digest** From neurons to sperm, our bodies are formed of a range of cells tailored to perform a unique role. However, organisms also host small reservoirs of unspecialized 'stem cells' that retain the ability to become different kinds of cells. When these stem cells divide, one daughter cell remains a stem cell while the other undergoes a series of changes that allows it to mature into a specific cell type.

This 'differentiation' process involves quickly switching off the stem cell programme, the set of genes that give a cell the ability to keep dividing while maintaining an unspecialized state. Failure to do so can result in the differentiating cell reverting towards its initial state and multiplying uncontrollably, which can lead to tumours and other health problems. While scientists have a good understanding of how the stem cell programme is turned off during differentiation, controlling these genes is a balancing act that starts even before division: if the program is over-active in the 'mother' stem cell, for instance, the systems that switch it off in its daughter can become overwhelmed. The mechanisms presiding over these steps are less well-understood.

To address this knowledge gap, Rajan, Anhezini et al. set out to determine how stem cells present in the brains of fruit flies could control the level of activity of their own stem cell programme. RNA sequencing and other genetic analyses revealed that a protein unique to these cells, called Fruitless, was responsible for decreasing the activity of the programme.

Biochemical experiments then showed that Fruitless performed this role by attaching a small amount of chemical modifications (called methyl groups) to the proteins that 'package' the DNA near genes involved in the stem cell programme. High levels of methyl groups present near a gene will switch off this sequence completely; however, the amount of methyl groups that Fruitless helped to deposit is multiple folds lower. Consequently, Fruitless 'fine-tunes' the activity of the stem cell programme instead, dampening it just enough to stop it from overpowering the 'off' mechanism that would take place later in the daughter cell.

These results shed new light on how stem cells behave – and how our bodies stop them from proliferating uncontrollably. In the future, Rajan, Anhezini et al. hope that this work will help to understand and treat diseases caused by defective stem cell differentiation.

state in stem cells can overwhelm translational control that downregulates their activity in stem cell progeny and perturb timely onset of differentiation (*San-Juán and Baonza, 2011*; *Xiao et al., 2012*; *Zacharioudaki et al., 2012*; *Zhu et al., 2012*; *Larson et al., 2021*; *Ohtsuka and Kageyama, 2021*). Conversely, excessive transcription of differentiation genes that instill biases toward terminal cellular functions in stem cells can overcome the mechanisms that uncouple these transcripts from the translational machinery and prematurely deplete the stem cell pool (*Lennox et al., 2018*; *Baser et al., 2019*; *de Rooij et al., 2019*; *Marques et al., 2023*). Thus, fine-tuning stemness and differentiation gene transcription in stem cells minimizes inappropriate gene activity that could result in developmental anomalies. Coordinated regulation of stemness and differentiation gene activity in stem cells by transcriptional and translational control poise stem cell progeny for a timely cell-state transition during differentiation (*Ables et al., 2011*; *Koch et al., 2013*; *Kobayashi and Kageyama, 2014*; *Bigas and Porcheri, 2018*; *Rajan et al., 2021*). Mechanistic investigation of the fine-tuning of stemness and differentiation gene transcription in vivo is challenging due to the compensatory effect of translational control, a lack of sensitized functional readouts, and a lack of insight into relevant transcription factors.

Neuroblast lineages of the fly larval brain provide an excellent in vivo paradigm for mechanistic investigation of gene regulation during developmental transitions because the cell-type hierarchy is well-characterized at functional and molecular levels (*Janssens and Lee, 2014*; *Homem et al., 2015*; *Doe, 2017*). A larval brain lobe contains approximately 100 neuroblasts, and each neuroblast asymmetrically divides every 60–90 min, regenerating itself and producing a sibling progeny that commits to generating differentiated cell types. Most of these neuroblasts are type I, which generate a ganglion mother cell (GMC) in every division. A GMC undergoes terminal division to produce two neurons. Eight neuroblasts are type II, which invariably generate an immature intermediate neural progenitor (immature INP) in every division (*Bello et al., 2008*; *Boone and Doe, 2008*; *Bowman et al., 2008*). An immature INP initiates INP commitment 60 min after asymmetric neuroblast division

(*Janssens et al., 2017*). An immature INP initially lacks Asense (Ase) protein expression and upregulates Ase as it progresses through INP commitment. Once INP commitment is complete, an Ase⁺ immature INP transitions into an INP and asymmetrically divides 5–6 times to generate more than a dozen differentiated cells, including neurons and glia (*Viktorin et al., 2011*; *Bayraktar and Doe, 2013*). All type II neuroblast lineage cell types in larval brains can be unambiguously identified based on functional characteristics and protein marker expression. Single-cell RNA-sequencing (scRNA-seq) of sorted, fluorescently labeled INPs and their differentiating progeny from wild-type brain tissue has led to the discovery of many new genes that contribute to the generation of diverse differentiated cell types during neurogenesis (*Michki et al., 2021*). This wealth of information on the type II neuroblast lineage allows for mechanistic investigations of precise spatiotemporal regulation of gene expression during developmental transitions.

A multilayered gene regulation system ensures timely onset of INP commitment in immature INPs by coordinately terminating Notch signaling activity (*Komori et al., 2018*). Activated Notch signaling drives the expression of downstream-effector genes *deadpan (dpn)* and *Enhancer of (splitz)mγ (E(spl) mγ)*, which promote stemness in type II neuroblasts by poising activation of the master regulator of INP commitment *earmuff (erm)* (*San-Juán and Baonza, 2011*; *Xiao et al., 2012*; *Zacharioudaki et al., 2012*; *Zhu et al., 2012*; *Zacharioudaki et al., 2016*). During asymmetric neuroblast division, the basal protein Numb and Brain tumor (Brat) exclusively segregate into immature INPs, where they terminate Notch signaling activity and promote the timely onset of Erm expression (*Bello et al., 2006*; *Betschinger et al., 2006*; *Lee et al., 2006a*; *Lee et al., 2006b*; *Wang et al., 2006*). Numb is a conserved Notch inhibitor and prevents continual Notch activation in immature INPs (*Frise et al., 1996*; *Zhong et al., 1996*; *Lee et al., 2006a*; *Wang et al., 2006*; *Wirtz-Peitz et al., 2008*). Asymmetric segregation of the RNA-binding protein Brat is facilitated by its adapter protein Miranda, which releases Brat from the cortex of immature INPs, allowing Brat to promote decay of Notch downstream-effector gene transcripts and thus initiate differentiation (*Loedige et al., 2014*; *Laver et al., 2015*; *Loedige et al., 2015*; *Komori et al., 2018*; *Reichardt et al., 2018*). Complete loss of *numb* or *brat* function leads to unrestrained activation of Notch signaling in immature INPs driving them to revert into type II neuroblasts leading to a severe supernumerary neuroblast phenotype. Similarly, increased levels of activated Notch or Notch transcriptional target gene expression in immature INPs can drastically enhance the moderate supernumerary neuroblast phenotype in *brat-* or *numb*-hypomorphic brains (*Xiao et al., 2012*; *Janssens et al., 2014*; *Komori et al., 2014b*; *Komori et al., 2018*; *Larson et al., 2021*). Collectively, these findings suggest that precise transcriptional control of *Notch* and Notch target gene expression levels during asymmetric neuroblast division is essential, safeguarding the generation of neurons that are required for neuronal circuit formation in adult brains.

We defined the fine-tuning of stemness gene transcription as a function that is mild enough to not effect INP commitment when lost alone but enough to enhance immature INP reversion to supernumerary neuroblasts induced by decreased post-transcriptional control of stemness gene expression. We established three key criteria to identify regulators which fine-tune stemness gene transcription in neuroblasts, (1) an established role in transcriptional regulation, for example a DNA-binding transcription factor, (2) clear expression in neuroblasts with no protein expression in immature INPs, and (3) acts as a negative regulator of its targets. From a type II neuroblast lineage-specific single-cell gene transcriptomic atlas, we found that *fruitless (fru)* mRNAs are detected in type I & II neuroblasts but not in their differentiating progeny. One specific Zn-finger containing isoform of Fru (Fruᶜ) is exclusively expressed in all neuroblasts. Fruᶜ binds *cis*-regulatory elements of most genes uniquely transcribed in type II neuroblasts, including *Notch* and Notch downstream-effector genes that promote stemness in neuroblasts. A modest increase in *Notch* or Notch downstream gene expression induced by loss of *fruᶜ* function alone has no effect on INP commitment, but enhances immature INP reversion to type II neuroblasts in *numb-* and *brat*-hypomorphic brains. To establish how Fruᶜ might fine-tune gene transcription in neuroblasts, we examined the distribution of established histone modifications in the presence or absence of *fruᶜ*. We surprisingly found Fruᶜ-dependent low-level enrichment of the repressive histone marker H3K27me3 in most Fruᶜ-bound peaks in genes uniquely transcribed in type II neuroblasts including *Notch* and its downstream-effector genes. The Polycomb Repressive Complex 2 (PRC2) subunits are enriched in Fruᶜ-bound peaks in genes uniquely transcribed in type II neuroblasts, and reduced PRC2 function enhances the supernumerary neuroblast phenotype in *numb*-hypomorphic brains, identical to *fruᶜ* loss-of-function. We conclude that the Fruᶜ-PRC2-H3K27me3

molecular pathway fine-tunes stemness gene expression in neuroblasts by promoting low-level H3K27me3 enrichment in their *cis*-regulatory elements. The mechanism by which PRC2-H3K27me3 fine-tune stem cell gene expression will likely be relevant throughout metazoans.

## Results

### A gene expression atlas captures dynamic changes throughout type II neuroblast lineages

To identify regulators of gene transcription during asymmetric neuroblast division, we constructed a single-cell gene transcription atlas that encompasses all cell types in the type II neuroblast lineage in larval brains. We fluorescently labeled all cell types in the lineage in wild-type third-instar larval brains, sorted positively labeled cells by flow cytometry, and performed single-cell RNA-sequencing (scRNA-seq) using a 10x genomic platform (*Figure 1A*; *Figure 1—figure supplement 1A*). This new dataset displays high levels of correlation to our previously published scRNA-seq dataset which were limited to INPs and their progeny. The harmonization of these two datasets results in a gene transcription atlas of the type II neuroblast lineage consisting of over 11,000 cells (*Figure 1B*). Based on the expression of known cell identity genes, we were able to observe clusters consisting of type II neuroblasts ($dpn^+,pnt^+$), INPs ($dpn^+,opa^+$), GMCs ($dap^+,hey^-$), immature neurons ($dap^+,hey^+$), mature neurons ($hey^-,nSyb^+$), and glia ($repo^+$) (*Figure 1C*). The UMAP positions of these clusters match well with the results of pseudo-time analyses from a starter cell that was positive for *dpn*, *pnt*, and *RFP* transcripts (*Figure 1D*). Leiden clustering of the data was able to capture these major cell types (*Figure 1E*), and quality control metrics showed most clusters captured on average 1.5 k genes and showed low mitochondrial gene expression (*Figure 1—figure supplement 1B*). Thus, the harmonized scRNA-seq dataset captures molecularly and functionally defined stages of differentiation in the type II neuroblast lineage (*Figure 1F*).

To determine whether the new scRNA-seq dataset encompasses neuroblast progeny undergoing dynamic changes in cell identity during differentiation, we examined transcripts that were transiently expressed in neuroblast progeny undergoing INP commitment or asymmetric INP division. We found that cluster 14 contains type II neuroblasts ($dpn^+,erm^-,ase^-,ham^-$), Ase⁻ immature INPs ($dpn^-,erm^+,ase^-,ham^-$) and Ase⁺ immature INPs ($dpn^-,erm^-,ase^+,ham^+$) (*Figure 1E*), which are well-defined rapidly changing transcriptional states during INP commitment (*Xiao et al., 2012*; *Janssens et al., 2014*; *Rives-Quinto et al., 2020*). Furthermore, cluster 1 contains proliferating INPs that express known differential temporal transcription factors (*Bayraktar and Doe, 2013*; *Tang et al., 2022*), including young INPs ($D^+,hbn^-,ey^-,scro^-$) and old INPs ($D^-,hbn^+,ey^+,scro^+$) (*Figure 1E*). These data led us to conclude that the type II neuroblast lineage gene transcription atlas captures neuroblast progeny undergoing dynamic changes in cell identity during differentiation (*Figure 1F*).

### Fru$^C$ negatively regulates stemness gene expression in neuroblasts

We hypothesized that regulators that fine-tune stemness gene expression in neuroblasts should (1) be transcription factors, (2) be exclusively expressed in type II neuroblasts, and (3) negatively regulate gene transcription. We searched for candidate genes that fulfill these criteria in the cluster 14 of the type II neuroblast lineage gene transcription atlas. *dpn* serves as a positive control because its transcripts are highly enriched in type II neuroblasts and rapidly degraded in Ase⁻ immature INPs, allowing us to distinguish neuroblasts from immature INPs (*Figure 2A–B*). We found the expression of *fru* mirrors *dpn* expression, with transcript levels high in type II neuroblasts but lower in Ase⁻ immature INPs (*Figure 2A*). *fru* is a pleiotropic gene with at least two major functions: one that controls male sexual behavior and another that is essential for viability in both sexes (*Goodwin and Hobert, 2021*). *fru* transcripts are alternatively spliced into multiple isoforms that encode putative transcription factors containing a common BTB (protein-protein interaction) N-terminal domain and one of four C-terminal zinc-finger DNA-binding domains (*Dalton et al., 2013*; *Neville et al., 2014*; *von Philipsborn et al., 2014*; *Figure 2A*). We used the Fru-common antibody that recognizes all isoforms to determine the spatial expression pattern of Fru protein in green fluorescent protein (GFP)-marked wild-type neuroblast clones. We detected Fru in neuroblasts but found that Fru is rapidly downregulated in their differentiating progeny in type I and II lineages (*Figure 2B*). To determine which Fru isoform is expressed in neuroblasts, we examined the expression of isoform-specific *fru::Myc* tagged

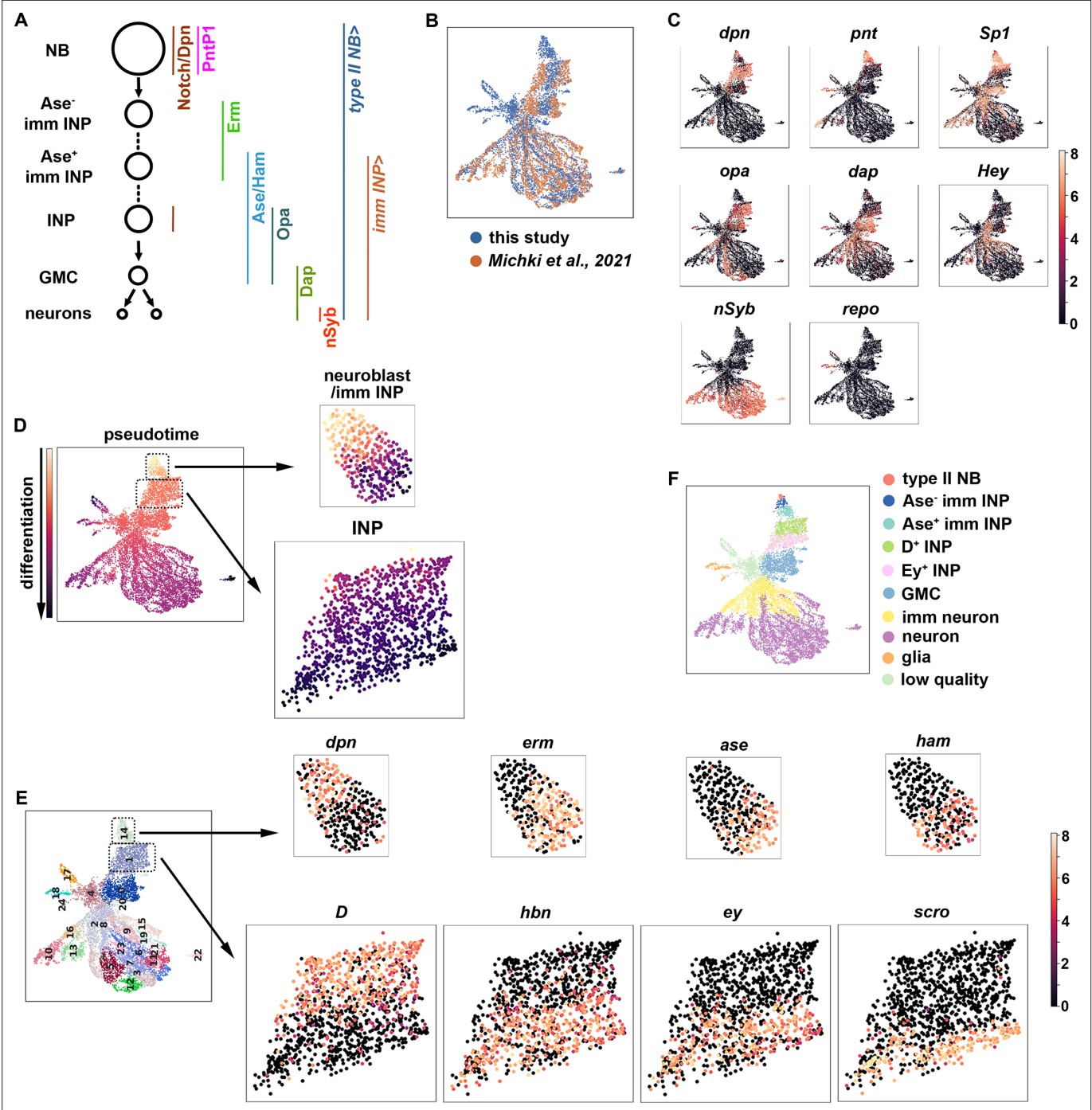

**Figure 1.** A single-cell gene expression atlas of type II neuroblast lineages. (**A**) Summary illustration of gene and Gal4 driver expression patterns in the type II neuroblast lineage. The type II NB Gal4 driver: *Wor-Gal4,Ase-Gal80*. imm INP driver: *R9D11-Gal4*. (**B**) Harmonization of the scRNA-seq dataset from the entire type II neuroblast (NB) lineage generated in this study (blue) and our previously published scRNA-seq dataset which were limited to INPs and their progeny (orange). The genotype of larval brains used for scRNA-seq in this study: *UAS-dcr2; Wor-Gal4, Ase-Gal80; UAS-RFP::stinger*. (**C**) UMAPs of known cell-type-specific marker genes. Color intensity indicated scaled (log1p) gene expression value. (**D**) Pseudotime analysis starting from cells enriched for *dpn*, *pnt*, and *RFP* transcripts. (**E**) Left: Leiden clustering of the scRNA-seq atlas. Right: Representative UMAPs of dynamically expressed transcription factors from clusters 14 (NBs and immature INPs) and 1 (INPs). Color intensity indicated scaled (log1p) gene expression value. (**F**) Annotated gene expression atlas of a wild-type type II neuroblast lineage.

The online version of this article includes the following figure supplement(s) for figure 1:

**Figure supplement 1.** Quality control data for the scRNA-seq atlas.

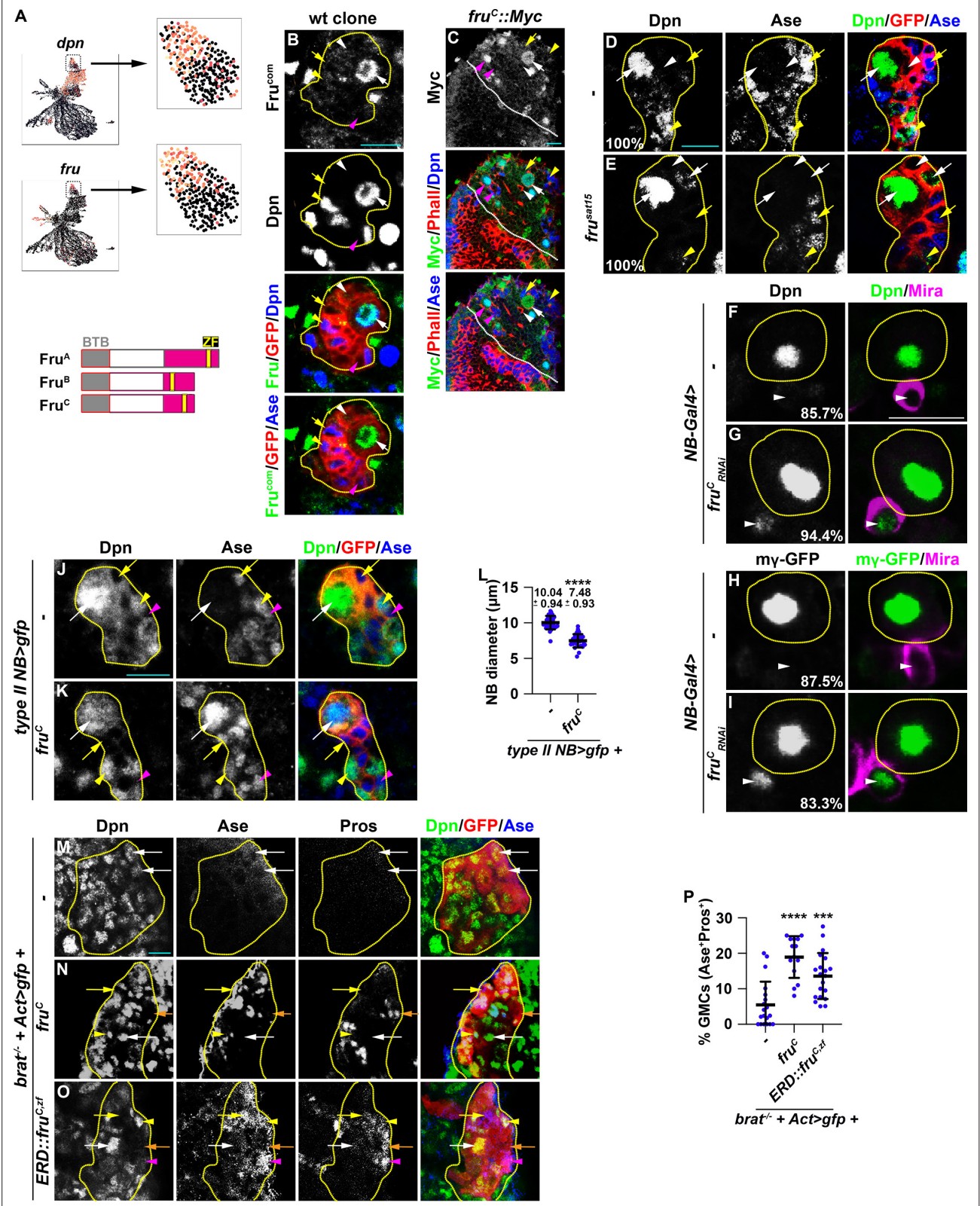

**Figure 2.** Fru[c] functions through transcriptional repression to regulate stemness gene expression. (**A**) Top: *fru* and *dpn* mRNAs are highly enriched in neuroblasts in cluster 14 of the scRNA-seq dataset, but only *dpn* mRNAs are detected in INPs in cluster 1. Bottom: Domains in Fru protein isoforms. ZF: zinc-finger DNA-binding domain. (**B**) Fru protein is detected in the neuroblast but not in INPs in a GFP-marked type II neuroblast lineage clone. The genotype used in this experiment is *Elav-Gal4,UAS-mCD8::GFP,hs-flp; FRT82B,Tub-Gal80/FRT82B*. (**C**) Endogenously expressed Fru[c] tagged by

*Figure 2 continued on next page*

*Figure 2 continued*

a Myc epitope (*fru^C^::Myc*) is detected in type I and II neuroblasts but not in their differentiating progeny. (**D–E**) Immature INPs in GFP-marked wild-type type II neuroblast clones never show detectable Dpn expression (100%). 1–2 Ase⁻ immature INP per *fru*-null (*fru^sat15^*) type II neuroblast clone show detectable Dpn expression (100%). The genotype in this experiment is *Elav-Gal4,UAS-mCD8::GFP,hs-flp; FRT82B,Tub-Gal80/FRT82B,fru^Sat15^*. (**F–I**) The newborn immature INP marked by cortical Miranda staining show undetectable Dpn and E(spl)mγ::GFP expression in the majority of wild-type clones (85.7%). Reducing *fru^C^* function leads to ectopic Dpn and E(spl)mγ::GFP expression in the newborn immature INP in most type II neuroblast lineages (94.4%). NB-Gal4: *Wor-Gal4*. (**J–L**) Type II neuroblasts overexpressing Fru^C^ display characteristics of differentiation including a reduced cell diameter and aberrant Ase expression. Quantification of the cell diameter is shown in L. (*type II NB>: Wor-Gal4, Ase-Gal80*). (**M–P**) Overexpressing full-length Fru^C^ or a constitutive transcriptional repressor form of Fru^C^ (Fru^c,zf^::ERD) is sufficient to partially restore differentiation in *brat*-null type II neuroblast clones (*brat^11^/ Df(2 L)Exel8040,hs-flp; Act5C-Gal4>FRT > FRT>UAS-GFP/UAS-fru^C^* or *ERD::fru^C,zf^*). The percentage of GMCs per clone is shown in P. Yellow dashed line encircles a type II neuroblast lineage. White dotted line separates optic lobe from brain. white arrow: type II neuroblast; white arrowhead: Ase⁻ immature INP; yellow arrow: Ase⁺ immature INP; yellow arrowhead: INP; magenta arrow: type I neuroblast; magenta arrowhead: GMC. Scale bars: 10 µm. p-values: ***<0.0005, and ****<0.00005.

The online version of this article includes the following figure supplement(s) for figure 2:

**Figure supplement 1.** Fru^A^ and Fru^B^ are ubiquitously expressed in neuroblast lineages.

allele where a Myc epitope is knocked into the C-terminus of the Fru^A^, Fru^B^, or Fru^C^ coding region (***von Philipsborn et al., 2014***). While Fru^A^::Myc and Fru^B^::Myc appear to be ubiquitously expressed at low levels, Fru^C^::Myc is specifically expressed in both types of neuroblasts but not in their differentiating progeny, including immature INPs, INPs, and GMCs (***Figure 2C***, ***Figure 2—figure supplement 1A–B***). These data indicate that Fru^C^ is the predominant Fru isoform expressed in neuroblasts.

To define the function of Fru in neuroblasts, we assessed the identity of cells in the GFP-marked mosaic clones derived from single type II neuroblasts. The wild-type neuroblast clone always contains a single neuroblast that can be uniquely identified by cell size (10–12 µm in diameter) and marker expression (Dpn⁺Ase⁻) as well as 6–8 smaller, Dpn⁻ immature INPs (***Figure 2D***). The neuroblast clone carrying deletion of the *fru* locus (*fru^-/-^*) contains a single identifiable neuroblast but frequently contains multiple Ase⁻ immature INPs with detectable Dpn expression (***Figure 2E***). Over 80% of newborn immature INPs (marked by intense cortical Mira expression) generated by *fru^C^*-mutant type II neuroblasts ectopically express Dpn and E(spl)mγ while less than 15% of newborn immature INPs generated by wild-type neuroblasts expressed these genes (***Figure 2F–I***). These results support a model in which loss of *fru^C^* function increases the expression of Notch downstream-effector genes that promote stemness in neuroblasts. Consistently, type II neuroblasts overexpressing Fru^C^ prematurely initiate INP commitment, as indicated by a reduced cell diameter and precocious Ase expression (***Figure 2J–L***). Thus, loss of *fru^C^* function increases stemness gene expression whereas gain of *fru^C^* decreases stemness gene expression during asymmetric neuroblast division.

To determine whether Fru^C^ negatively regulates the transcription of stemness genes in type II neuroblasts, we overexpressed wild-type Fru^C^ in GFP-marked neuroblast lineage clones in *brat*-null brains. Ectopic translation of Notch downstream-effector gene transcripts that promote stemness in neuroblasts drives immature INP reversion to supernumerary type II neuroblasts at the expense of differentiating cell types in *brat*-null brains (***Loedige et al., 2015***; ***Komori et al., 2018***; ***Reichardt et al., 2018***). Control clones in *brat*-null brains contain mostly type II neuroblasts and few differentiating cells that include Ase⁺ immature INPs, INPs, and GMCs (***Figure 2M and P***). By contrast, overexpressing full-length Fru^C^ increases the number of INPs, GMCs, and differentiating neurons (Ase⁻ Pros⁺) in *brat*-null neuroblast clones (***Figure 2N and P***). This result indicates that Fru^C^ overexpression is sufficient to partially restore differentiation in *brat*-null brains. To test if Fru^c^ restores differentiation by promoting transcriptional repression, we generated fly lines carrying the *UAS-fru^C,zf^::ERD* transgene that encodes the zinc-finger DNA-binding motif of Fru^C^ fused in frame with the Engrail Repressor Domain. The ERD domain is well conserved in multiple classes of homeodomain proteins as well as many transcriptional repressors across the bilaterian divide and binds to the Groucho co-repressor protein to exert its repressor function (***Smith and Jaynes, 1996***; ***Jiménez et al., 1997***; ***Bürglin and Affolter, 2016***). Several previously published studies have used this strategy to demonstrate that neurogenetic transcription factors exert transcriptional repression function in neuroblasts (***Xiao et al., 2012***; ***Janssens et al., 2014***; ***Bahrampour et al., 2017***; ***Rives-Quinto et al., 2020***). Similar to full-length Fru^C^ overexpression, overexpressing Fru^C,zf^::ERD was also sufficient to partially restore

differentiation in *brat*-null brains (**Figure 2O–P**). Thus, we conclude that Fru^C negatively regulates stemness gene expression in type II neuroblasts.

## Fru^c binds *cis*-regulatory elements of the majority of genes uniquely transcribed in neuroblasts

If Fru^C directly represses stemness gene expression, Fru^C should bind their *cis*-regulatory elements. To identify Fru^C-bound regions in neuroblasts, we applied a protocol of Cleavage Under Targets and Release Using Nuclease (CUT&RUN) to brain extracts from dissected third-instar *brat*-null larvae homozygous for the *fru^C::Myc* knock-in allele. *brat*-null brains accumulate thousands of supernumerary type II neuroblasts at the expense of INPs and provide a biologically relevant source of type II neuroblast-specific chromatin (**Komori et al., 2014a**; **Janssens et al., 2017**; **Komori et al., 2018**; **Rives-Quinto et al., 2020**; **Larson et al., 2021**). We used a specific antibody against the Myc epitope or the Fru-common antibody to confirm that Fru^C::Myc is detected in all supernumerary type II neuroblasts in *brat*-null brains homozygous for *fru^C::Myc* (**Figure 3—figure supplement 1A**). We determined the genome-wide occupancy of Fru^C::Myc in type II neuroblasts using the Myc antibody and Fru-common antibody, and found that Fru^C::Myc binding patterns revealed by these two antibodies are highly correlated (**Figure 3—figure supplement 1B**; Pearson correlation = 0.94). Fru^C binds 9301 regions in type II neuroblasts (**Figure 3A**). Overall, 59% of Fru^C-bound regions are promoters whereas 29% are enhancers in the intergenic and intronic regions (**Figure 3A**). By contrast, 15% of randomized control regions are promoters and 55% are enhancers (**Figure 3A**). 50.1% of Fru^C-bound regions in promoters and enhancers overlap with regions of accessible chromatin (**Larson et al., 2021**; **Figure 3B**). Consistent with the finding that Fru^C negatively regulates stemness gene expression, Fru^C binds promoters and neuroblast-specific enhancers of *Notch*, *dpn*, *E(spl)mγ*, *klumpfuss* (*klu*) and *tailless* (*tll*) that were previously shown to maintain type II neuroblasts in an undifferentiated state (**Figure 3C**; **Figure 3—figure supplement 1C**). Based on our scRNA-seq data, we classified genes as NB genes, imm INP genes, or invariant genes expressed throughout the lineage based on differential expression within cluster 14 (**Figure 1E**; **Supplementary file 2**). Seventy-four percent of genes uniquely transcribed in type II neuroblasts (NB genes) are bound by Fru^C whereas 41% of these genes are in randomized control (**Figure 3D–E**). By contrast, the percentage of Fru^C-bound genes transcribed in immature INPs or throughout the type II neuroblast lineage is similar to random control (**Figure 3D–E**). Because stemness gene transcripts are highly enriched in type II neuroblasts, these results suggest that Fru^C preferentially binds *cis*-regulatory elements of stemness genes.

A mechanism by which Fru^C can negatively regulate stemness gene expression levels is to modulate the activity of the Notch transcriptional activator complex activity. Using Affymetrix GeneChip, a previous study demonstrated that the Notch transcriptional activator complex binds 595 regions in 185 transcribed genes in neuroblasts ($\log_2$ FC >0.5) including *dpn*, *E(spl)mγ*, *klu*, and *tll* (**Zacharioudaki et al., 2016**). To precisely identify Notch-bound peaks in type II neuroblasts, we used a specific antibody against the DNA-binding subunit of the Notch transcriptional activator complex, Suppressor of Hairless (Su(H)), to perform a CUT&RUN assay on brain lysate form third-instar *brat*-null larvae homozygous for *fru^C::Myc*. Prior to the genomic study, we validated the specificity of the Su(H) antibody in vivo by performing immunofluorescent staining of *brat*-null larvae homozygous for *fru^C::Myc*. We observed co-expression of Su(H) and Dpn in thousands of supernumerary type II neuroblasts in *brat*-null brains (**Figure 3—figure supplement 1D**). Because *dpn* is directly activated by Notch in many cell types, including neuroblasts, this result confirms that the Su(H) antibody can detect activated Notch activity. We found that Su(H) binds 305 regions in 112 genes in type II neuroblasts, and that Su(H)-bound regions are predominantly in promoters and enhancers (**Figure 3F**). Twelve percent of NB genes are bound by Su(H) compared to 0% of these gene in a randomized control (**Figure 3D–E**). By contrast, the percentage of Su(H) bound genes transcribed in immature INPs or throughout the type II neuroblast lineage are similar to random control. Overall, 95.2% of Su(H)-bound promoters and 90.8% of Su(H)-bound enhancers overlap with Fru^C-bound peaks (**Figure 3G**). These peaks include the promoters of *Notch*, *dpn*, *E(spl)mγ*, *klu* and *tll* as well as the enhancers that drive their expression in neuroblasts (**Figure 3C**; **Figure 3—figure supplement 1C**). Our data support a model that Fru^C regulates Notch pathway gene expression by occupying functionally relevant regulatory elements bound by Notch in type II neuroblasts.

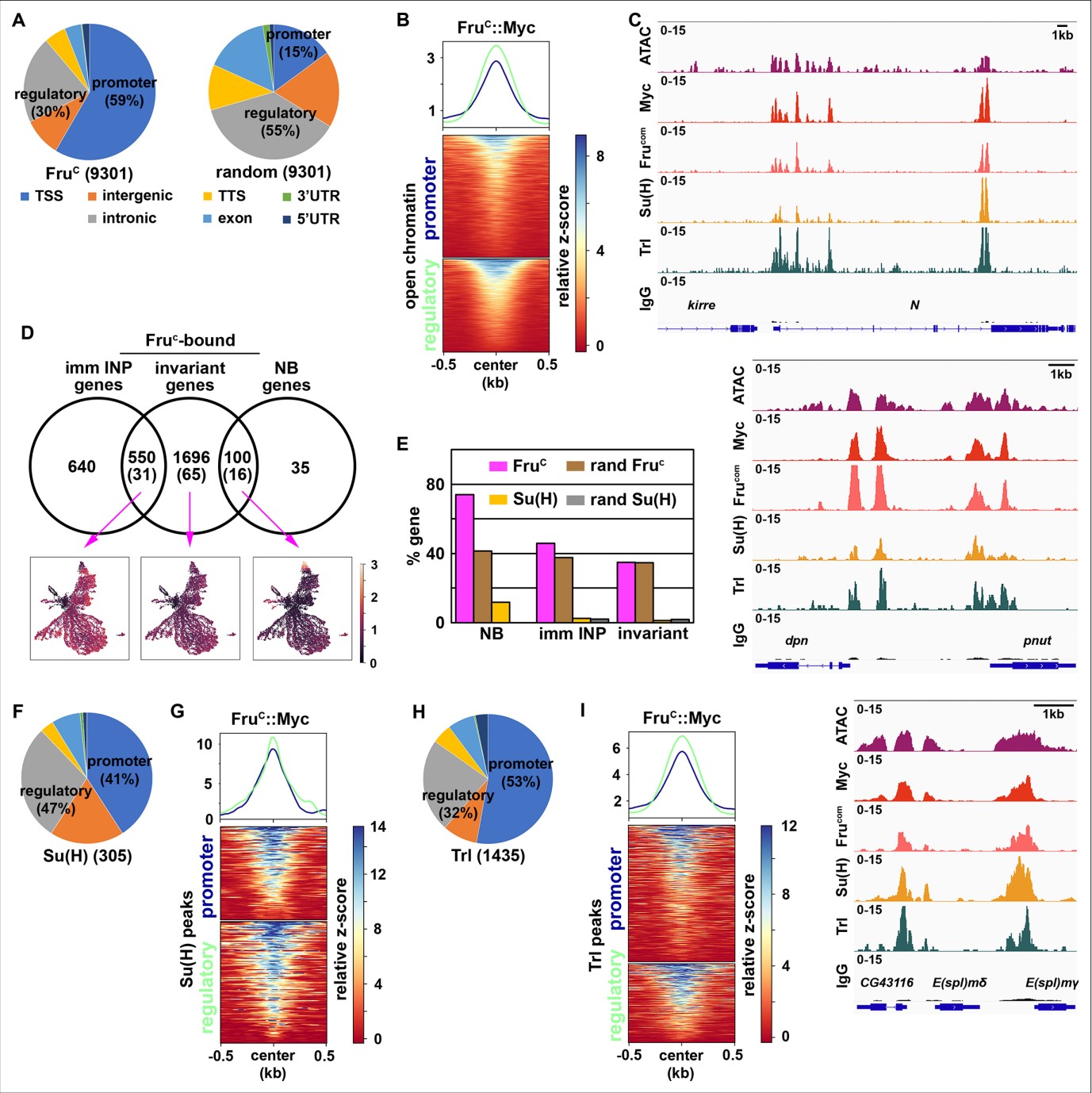

**Figure 3.** Fru$^c$ preferentially binds regulatory elements of genes uniquely expressed in neuroblasts. (**A**) Genomic binding distribution of Fru$^C$-bound peaks (total # of peaks shown in parentheses) from CUT&RUN or random (set of Fru$^c$ peaks shuffled to randomly determined places in the genome) in type II neuroblast-enriched chromatin from *brat*-null brains (*brat$^{11/Df(2L)Exel8040}$*). Fru$^c$ preferentially binds promoters. (**B**) Heatmap is centered on promoters or regulatory regions showing accessible chromatin as defined by ATAC-seq with 500 bp flanking regions and ordered by signal intensity of Fru$^c$::Myc binding. (**C**) Representative z score-normalized genome browser tracks showing regions with accessible chromatin (ATAC-seq) and bound by Fru$^c$::Myc (detected by the Myc antibody or Fru$^{COM}$ antibody), Su(H), Trl, or IgG at *Notch, dpn,* and *E(spl)mγ* loci. (**D**) Genes in cluster 14 from the scRNA-seq dataset were separated into neuroblast-enriched genes (right), immature INP-enriched genes (left), and invariant genes. The middle circle is the set of genes bound by Fru$^c$ or Su(H) (shown in parentheses). UMAPs show gene enrichment score for Fru$^c$-bound genes uniquely expressed in neuroblasts (right), uniquely expressed in immature INPs (left), and ubiquitously expressed (invariant genes) (middle). (**E**) Percentage of genes defined in D bound by either Fru$^c$, Su(H), random Fru$^c$ peaks, or random Su(H) peaks (set of Su(H) peaks shuffled to randomly determined places in the genome). (**F–G**) Genomic binding distribution of identified Su(H)-bound peaks (total # of peaks shown in parentheses) from CUT&RUN in type II neuroblast-enriched chromatin. Heatmap is centered on promoters or regulatory regions and ordered by signal intensity of Su(H) binding. (**H–I**) Genomic binding distribution

Figure 3 continued

of identified Trl-bound peaks (total # of peaks shown in parentheses) from CUT&RUN in type II neuroblast-enriched chromatin. Heatmap is centered on promoters or regulatory regions and ordered by signal intensity of Trl binding.

The online version of this article includes the following figure supplement(s) for figure 3:

**Figure supplement 1.** Quality control for determining Fru$^C$-, Su(H)- and Trl-binding using CUT&RUN.

De novo motif discovery identified a sequence bound by the transcription factor Trithorax-like (Trl), also known as GAGA factor, is significantly enriched in both Fru$^C$-bound promoters and enhancers (*Figure 3—figure supplement 1E*). The Trl motif was previously found to be the most significantly enriched in Fru$^C$-associated genomic regions in the larval nervous system (*Neville et al., 2014*). Trl, like Fru, is a member of the BTB-Zn-finger transcription factor family which heterodimerize with other BTB-domain-containing factors to regulate gene transcription (*Bonchuk et al., 2022*). Trl is an evolutionarily conserved multifaceted transcription factor that regulates diverse biological processes by interacting with a wide variety of proteins including PRC2 complex components (*Lomaev et al., 2017*; *Chetverina et al., 2021*; *Srivastava et al., 2018*). Although the Trl motif is generally enriched at promoters, enrichment of this motif in Fru$^C$-bound promoters and enhancers suggest that FruC might function together with Trl to negatively regulate gene transcription in type II neuroblasts. We examined whether Trl indeed binds Fru$^C$-bound regions in type II neuroblasts using a specific antibody against Trl (*Judd et al., 2021*). We validated the specificity of the Trl antibody in vivo by performing immunofluorescent staining of *brat*-null larvae homozygous for *fru$^C$::Myc*. We found that Trl is highly enriched in the nuclei of thousands of supernumerary type II neuroblasts marked by Dpn expression (*Figure 3—figure supplement 1F*). We used the Trl antibody to perform a CUT&RUN assay on brain lysate from third-instar *brat*-null larvae homozygous for *fru$^C$::Myc*. We identified 1435 Trl-bound regions, including promoters and enhancers, in type II neuroblasts (*Figure 3H*). In total, 85.4% of Trl-bound promoters and 87.8% of Trl-bound enhancers overlapped with Fru$^C$-bound regions including *Notch*, *dpn*, *E(spl)mγ*, *klu* and *tll* loci (*Figure 3C and I*; *Figure 3—figure supplement 1C*). These data suggest that Fru$^C$ may function together with Trl to regulate the transcription of stemness genes in neuroblasts.

## Fru$^C$ fine-tunes the transcription of Notch pathway genes during asymmetric neuroblast division

Loss- and gain-of-function of *fru$^C$* mildly alters the expression of Notch downstream-effector genes that promote stemness in neuroblasts (*Figure 2*). Thus, Fru$^C$ likely fine-tunes Notch signaling activity levels that balance neuroblast maintenance and INP commitment during neuroblast asymmetric division. To functionally validate the role of Fru$^C$ in fine-tuning Notch pathway activity expression, we tested whether loss of *fru$^C$* function can enhance the supernumerary neuroblast phenotype in *brat*-hypomorphic (*brat$^{hypo}$*) brains. Immature INPs revert to supernumerary type II neuroblasts at low frequency due to a modest increase in Notch downstream-effector gene expression in *brat$^{hypo}$* brains (*Komori et al., 2018*). Consistent with the finding that reduced *fru* function increases Notch downstream-effector protein levels in immature INPs, the heterozygosity of a *fru* deletion (*fru$^{-/+}$*) enhances the supernumerary neuroblast phenotype in *brat$^{hypo}$* brains (*Figure 4A*; *Figure 4—figure supplement 1A–B*). Furthermore, *brat$^{hypo}$* brains lacking *fru$^C$* function displayed greater than a twofold increase in supernumerary neuroblasts compared with *brat$^{hypo}$* brains heterozygous for a *fru* deletion (*Figure 4A*; *Figure 4—figure supplement 1C–E*). These data support our model that loss of *fru$^C$* function increases Notch pathway activity levels during asymmetric neuroblast division.

Because Fru$^C$ occupies enhancers relevant to the cell-type-specific expression of *Notch* and Notch downstream-effector genes in type II neuroblasts (*Figure 3*), we assessed whether Fru$^C$ fine-tunes the expression of Notch signaling pathway components during asymmetric neuroblast division. Proteolytic cleavage of the extracellular domain and the transmembrane fragment releases the Notch intracellular domain to form a transcriptional activator complex by binding Su(H) and Mastermind (Mam) (*Bray and Gomez-Lamarca, 2018*). Asymmetric segregation of Numb into immature INPs inhibits continual Notch activation and terminates Notch-activated transcription of its downstream-effector genes. *numb*-hypomorphic (*numb$^{hypo}$*) animals carrying the *numb$^{Ex}$* allele in trans with a *numb*-null allele (*numb$^{15}$*) contain more than 100 type II neuroblasts per brain lobe compared with 8 per lobe in

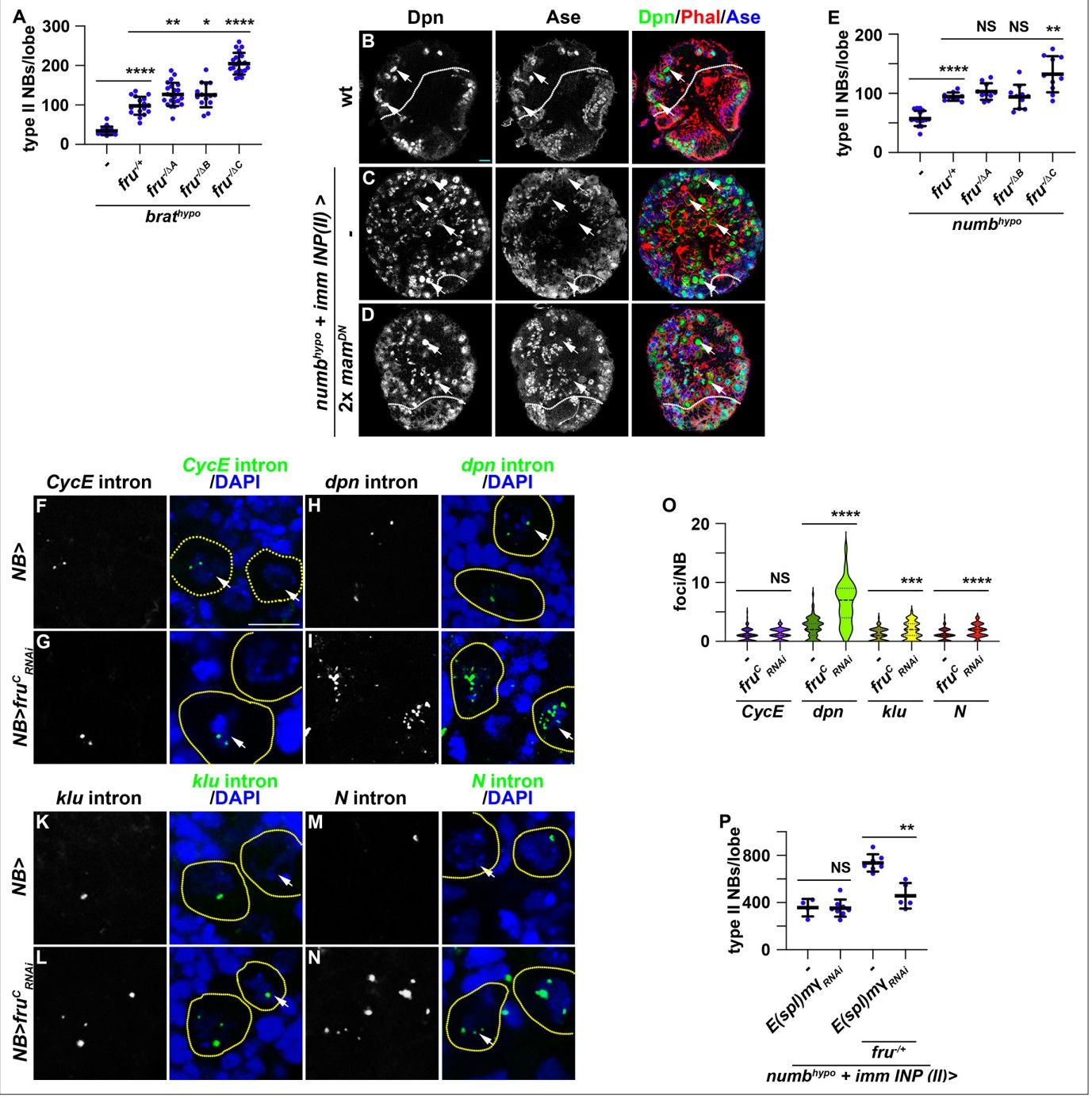

**Figure 4.** Reduced *fru* function increases Notch pathway gene transcription. (**A**) Loss of *fru^C* function (*fru^{ΔC/Aj96u3}*) enhances the supernumerary neuroblast phenotype in *brat^{hypo}* (*brat^{DG19310/11}*) brains heterozygous for a *fru* deletion (*fru^{Aj96u3/+}*), but loss of *fru^A* (*fru^{ΔA/Aj96u3}*) or *fru^B* (*fru^{ΔB/Aj96u3}*) function does not. (**B–D**) Overexpressing two copies of dominant-negative *mam* transgenes in immature INPs partially suppresses the supernumerary neuroblast phenotype in *numb^{hypo}* (*numb^{ex112/15}*) brains. The genotype used in this experiment is *numb^{Ex112}/R9D11-Gal4,numb^{15}* or *numb^{Ex112}/R9D11-Gal4,numb^{15};* *UAS-mam^{DN}/UAS-mam^{DN}*. (**E**) Loss of *fru^C* function (*fru^{ΔC/Aj96u3}*) enhances the supernumerary neuroblast phenotype in *numb^{hypo}* brains heterozygous for a *fru* deletion (*fru^{Aj96u3/+}*), but loss of *fru^A* (*fru^{ΔA/Aj96u3}*) or *fru^B* (*fru^{ΔB/Aj96u3}*) function does not. (**F–N**) sm-FISH using intron probes confirms increased *Notch* and Notch target gene (*dpn* and *klu*) transcription in *fru^C*-mutant neuroblasts comparing with control neuroblasts. *CycE* nascent transcripts serve as a control because *CycE* is not a Notch target gene, and *CycE* transcription is unaffected by *fru^c* knockdown. The genotype used in this experiments is *Wor-Gal4* or *Wor-Gal4/UAS-fru^C_{RNAi}* (**O**) Quantification of *Notch*, *dpn*, *klu* and *CycE* nascent transcript foci in control versus *fru^C*-mutant neuroblasts. sm-FISH signals were counted in 8 dorsal-most neuroblasts (>6 μm in diameter) per brain lobe. *CycE*: 1±0.97 (n=128 neuroblasts) in control; 1.16±0.91 (n=96 neuroblasts) in *fru^C_{RNAi}*. *dpn*: 2.33±1.65 (n=88 neuroblasts) in control; 6.52±3.55 (n=104 neuroblasts) in *fru^c_{RNAi}*. *klu*: 1.12±1.01 (n=80 neuroblasts) in

*Figure 4 continued on next page*

*Figure 4 continued*

control; 1.79±1.43 (n=112 neuroblasts) in *fru$^c_{RNAi}$*. *Notch*: 1.06±0.88 (n=112 neuroblasts) in control; 1.77±1.07 (n=104 neuroblasts) in *fru$^c_{RNAi}$*. (**P**) Knocking down *E(spl)mγ* function by *RNAi* in immature INPs suppresses increased supernumerary formation in *numb$^{hypo}$* brains heterozygous for *fru* while having no effect on *numb$^{hypo}$* brains alone. The genotypes used in this experiment are *numb$^{Ex112}$/R9D11-Gal4,numb$^{15}$; UAS-E(spl)mγ$_{RNAi}$/+* or *numb$^{Ex112}$/R9D11-Gal4,numb$^{15}$; UAS-E(spl)mγ$_{RNAi}$/fru$^{Aj96u3/+}$*. White dashed line separates brain from the optic lobe. Yellow dashed line encircles a neuroblast. white arrow: type II neuroblast. Scale bars: 10 µm. p-value: NS: non-significant, *<0.05, **<0.005, ***<0.0005, and ****<0.00005.

The online version of this article includes the following figure supplement(s) for figure 4:

**Figure supplement 1.** Loss of *fru$^C$* function enhances supernumerary neuroblast formation in *brat$^{hypo}$* and *numb$^{hypo}$* brains.

---

wild-type animals (*Figure 4B–C*; *Figure 4—figure supplement 1F–G*). Antagonizing Notch-activated gene transcription by overexpressing a dominant negative form of Mam (Mam$^{DN}$) in immature INPs suppressed the supernumerary neuroblast phenotype in *numb$^{hypo}$* brains (*Figure 4C–D*; *Figure 4—figure supplement 1G*). Thus, the supernumerary type II neuroblast phenotype in *numb$^{hypo}$* brains provides a direct functional readout of activated Notch levels during asymmetric neuroblast division. The heterozygosity of a *fru* deletion alone did not affect INP commitment in immature INPs but led to a twofold increase in supernumerary neuroblasts in *numb$^{hypo}$* brains (*Figure 4E*; *Figure 4—figure supplement 1H–I*). Complete loss of *fru$^C$* function (*fru$^{-/ΔC}$*) led to increased supernumerary neuroblast formation in *numb$^{hypo}$* brains compared with *fru$^{-/+}$* (*Figure 4E*; *Figure 4—figure supplement 1J–L*). These results suggest that loss of *fru$^C$* function increases Notch-activated gene expression in mitotic neuroblasts, and support a model that Fru$^C$ fine-tunes the transcription of Notch downstream-effector genes in neuroblasts.

To quantitatively evaluate whether Fru$^C$ fine-tunes *Notch* and Notch downstream-effector gene transcription in neuroblasts, we performed single-molecule fluorescent in-situ hybridization (sm-FISH) using intron probes to these transcripts in larval brains overexpressing a *fru$^C_{RNAi}$* transgene. Intron probes detect nascent transcripts allowing for quantitative measurement of gene transcription in the physiological context. Because *Cyclin E* (*CycE*) is not a Notch target, its nascent transcript levels should not be affected by alerted Notch signaling and serve as control in this experiment. Consistently, the number of *CycE* nascent transcript foci in neuroblasts appears statistically indistinguishable between control brains carrying only the *Gal4* driver or brains overexpressing *fru$^C_{RNAi}$* (*Figure 4F–G and O*). By contrast, the number of *Notch*, *dpn* and *klu* nascent transcript foci is significantly higher in *fru$^C$* knockdown brains than in control brains (*Figure 4H–O*). *E(spl)mγ* was exempted from this analysis because its open reading frame contains a single exon. These results strongly suggest that reduced *fru$^C$* function increases *Notch* and Notch downstream-effector gene transcription levels in neuroblasts. Consistent with this interpretation, knocking down the function of *E(spl)mγ* by RNAi strongly suppressed increased supernumerary neuroblast formation in *numb$^{hypo}$* brains heterozygous for *fru* while not effecting the baseline supernumerary neuroblast phenotype in *numb$^{hypo}$* brains (*Figure 4P*). The *dpn$_{RNAi}$* transgene was exempted from this analysis because of off-target effect. This result provides functional support of our model that reducing *fru$^C$* function increases Notch target gene transcription in neuroblasts. Thus, we conclude that Fru$^C$ fine-tunes *Notch* and Notch downstream-effector gene expression during asymmetric neuroblast division.

## Fru$^C$ fine-tunes gene expression by promoting low-level H3K27me3 enrichment

To define the mechanisms by which Fru$^C$ fine-tunes Notch pathway gene transcription in mitotic neuroblasts, we compared genome-wide patterns of histone marks by CUT&RUN in *brat*-null brains carrying a *fru$^{-/ΔC}$* allelic combination (*fru$^C$*-null) with that of *brat*-null brains alone (control) (*Figure 5—figure supplement 1A–B*). We used a 500 bp sliding window to search for regions that show changes in histone mark levels with a Q-value <0.05 in *fru$^C$*-null brains relative to control brains. We first focused on acetylated lysine 27 on histone H3 (H3K27ac) because a previous study suggested that Fru$^C$ functions through Histone deacetylase 1 (Hdac1) to regulate gene transcription during specification of sexually dimorphic neurons (*Ito et al., 2012*). If Fru$^C$ fine-tunes *Notch*, *dpn* and *E(spl)mγ* transcription in type II neuroblasts by promoting deacetylation of H3K27 at their promoters and neuroblast-specific enhancers, these loci should display higher H3K27ac levels in *fru$^C$*-null brains than in control brains. 14.2% (4139 kB / 29,065 kB) of the regions with statistically significant changes in H3K27ac levels genome-wide displayed greater than twofold increase in this histone mark in *fru$^C$*-null brains while

0.4% (130 kB / 29,065 kB) of these regions showed greater than twofold decrease (*Figure 5—figure supplement 1C*). 3.4% (306 kB / 8,870 kB) of Fru$^C$-bound regions showed greater than 2-fold increase in H3K27ac levels in *fru$^C$*-null brains, and these regions did not include the *cis*-regulatory elements and the bodies of *Notch*, *dpn* and *E(spl)mγ* (*Figure 5A*; *Figure 5—figure supplement 1D–E*). Thus, loss of *fru$^C$* function did not significantly increase the number of Fru$^C$-bound loci with greater than 2-fold increase in H3K27ac comparing with the genome overall. We conclude that H3K27 deacetylation likely plays a minor role in Fru$^C$-mediated fine-tuning of target gene transcription.

H3K4me3 is a chromatin mark associated with the promoters of actively transcribed genes (*Cenik and Shilatifard, 2021*). If Fru$^C$ fine-tunes *Notch*, *dpn* and *E(spl)mγ* transcription in type II neuroblasts by promoting demethylation of H3K4 at their promoters, these loci should display higher H3K4me3 levels in *fru$^C$*-null brains than in control brains. 26.7% (8788 kB / 32,939 kB) of the regions with statistically significant changes in H3K4me3 levels genome-wide displayed greater than twofold increase in *fru$^C$*-null brains while 4.2% (1391 kB / 32,939 kB) of these regions showed greater than 2-fold decrease (*Figure 5—figure supplement 1F*). 7.45% (831 kB / 11,149 kB) of Fru$^C$-bound regions showed greater than 2-fold increase in this histone mark in *fru$^C$*-null brains, and these regions did not include the promoters and the bodies of *Notch*, *dpn*, and *E(spl)mγ* (*Figure 5A*; *Figure 5—figure supplement 1D and G*). Thus, loss of *fru$^C$* function did not significantly increase the number of Fru$^C$-bound loci with greater than twofold increase in H3K4me3 comparing with the genome overall. We conclude that H3K4 demethylation unlikely plays a role in Fru$^C$-mediating fine-tuning of gene transcription.

High levels of H3K27me3 are associated with inactive enhancers and the body of repressed genes (*Laugesen et al., 2019*; *Piunti and Shilatifard, 2021*). Low H3K27me3 levels occur in active loci in human and mouse embryonic stem cells but are generally regarded as noise (*Mikkelsen et al., 2007*; *Pan et al., 2007*). If Fru$^C$ fine-tunes *Notch*, *dpn* and *E(spl)mγ* transcription in type II neuroblasts by promoting trimethylation of H3K27 at their neuroblast-specific enhancers, these loci should display lower H3K27me3 levels in *fru$^C$*-null brains than in control brains. High levels of H3K27me3 are deposited in broad domains that are frequently referred to as Polycomb domains (Pc domains) (*Brown et al., 2018*). Pc domains rarely overlapped with Fru$^C$-bound loci and did not appear to be disrupted in *fru$^C$*-null brains (*Figure 5B*). 5.7% (1429 kB / 24,888 kB) of the regions with statistically significant changes in H3K27me3 levels displayed greater than twofold increase in *fru$^C$*-null brains and 18.1% (4510 kB / 24,888 kB) of these regions showed greater than 2-fold decrease (*Figure 5C*). Importantly, 43.3% (2748 kB / 6,340 kB) of Fru$^C$-bound regions showed greater than twofold decrease in this histone mark in *fru$^C$*-null brains (*Figure 5D–E*; *Figure 5—figure supplement 1H*). Thus, loss of *fru$^C$* function significantly increases the number of Fru$^C$-bound loci with greater than twofold decrease in H3K27me3 comparing with the genome overall. We conclude that H3K27 trimethylation likely plays an important role in Fru$^C$-mediating fine-tuning of gene transcription.

H3K27me3 levels at Fru$^C$-bound peaks including *Notch* and Notch downstream-effector gene loci appear significantly lower than those in Pc domains in control brains suggesting that Fru$^C$ fine-tunes gene transcription by promoting low levels of H3K27me3 enrichment (*Figure 5A–B*; *Figure 5—figure supplement 1D*). To unbiasedly assess H3K27me3 levels at Fru$^C$-bound peaks, we compared the distribution of H3K27me3 reads in 500 bp bins throughout the genome versus Fru$^C$-bound peaks. The distribution of reads throughout the genome does not appreciably change between *fru$^C$*-null brains and control brains (*Figure 5F*). Consistent with FruC promoting H3K27me3 deposition at regions bound by FruC, there are many more bins that overlapped with Fru$^C$-bound peaks containing a reduced number of reads in *fru$^C$*-null brains comparing with control brains (*Figure 5F*). We compared the coverage (reads/bp) of H3K27me3 in Fru$^C$-bound peaks to canonically defined Pc domains in neuroblasts, and found that Fru$^c$ peaks had 3-fold less coverage of H3K27me3 than canonical Pc domains (*Figure 5G*). These results indicate that Fru$^C$ fine-tunes gene transcription by promoting low levels of H3K27me3 enrichment.

## PRC2 fine-tunes gene expression during asymmetric neuroblast division

PRC2 is thought to be the only enzymatic complex that catalyzes H3K27me3 deposition (*Laugesen et al., 2019*; *Piunti and Shilatifard, 2021*). If Fru$^C$ functions through low levels of H3K27me3 enrichment to finely tune *Notch*, *dpn*, and *E(spl)mγ* expression in mitotic neuroblasts, PRC2 core components should be enriched in Fru$^C$-bound peaks in type II neuroblasts and reducing PRC2 activity should enhance the supernumerary neuroblast phenotype in *numb$^{hypo}$* brains, identical to the result obtained

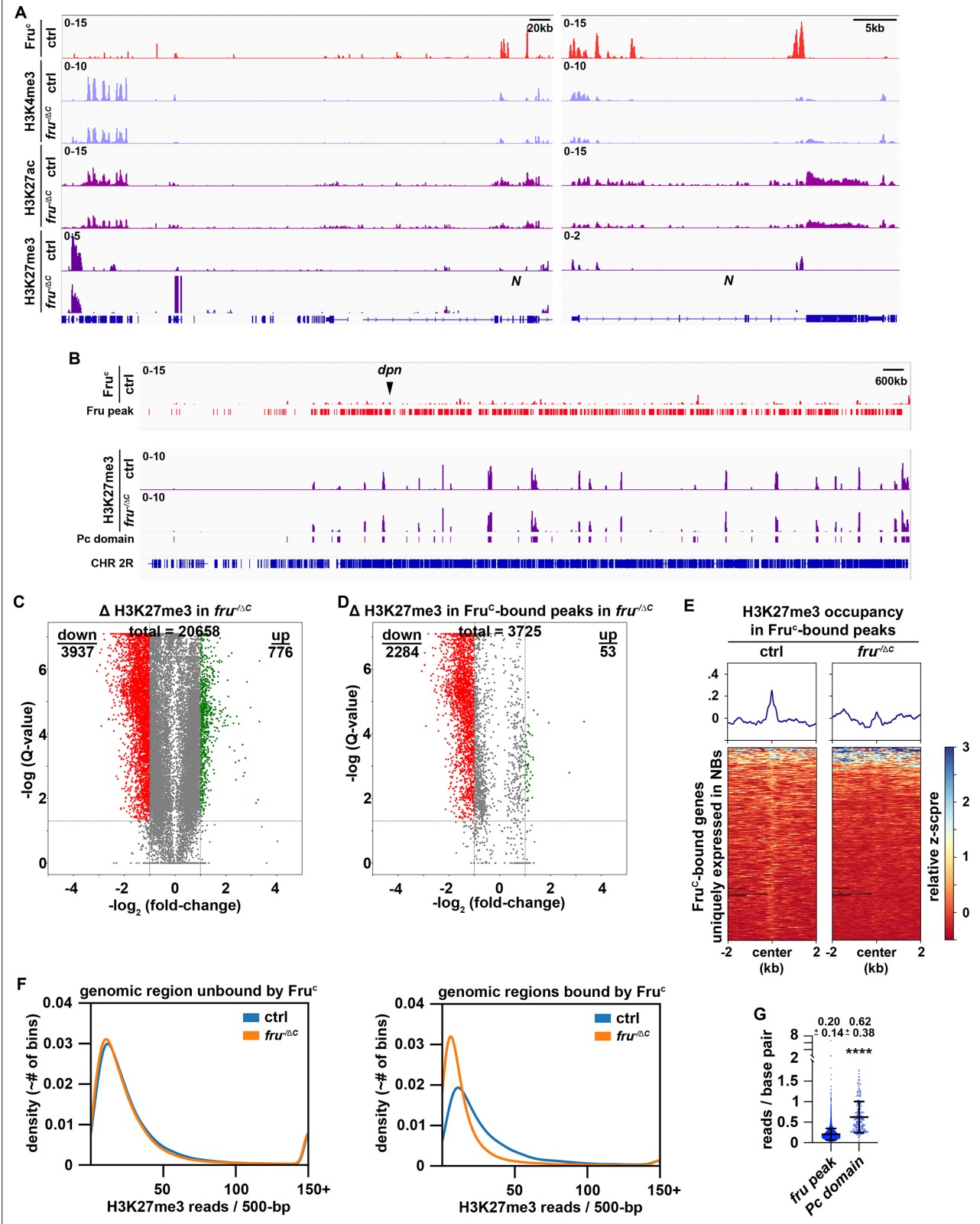

**Figure 5.** Low levels of H3K27me3 are enriched in Fru$^C$-bound regions. (**A**) Representative z score-normalized genome browser tracks showing Fru$^C$-binding and the enrichment of H3K4me3, H3K27ac, and H3K27me3 at the *Notch* locus in type II neuroblasts in control (*brat$^{11/Df(2L)Exel8040}$*) or fru$^C$-null (*brat$^{11/Df(2L)Exel8040}$; fru$^{\Delta C/Aj96u3}$*) brains. Left: Zoomed-out images showing nearest heterochromatin domains. Right: Zoomed-in images showing enrichment of histone marks in Fru$^C$-bound regions. (**B**) Representative z score-normalized genome browser track showing Fru$^C$-binding and the H3K27me3

*Figure 5 continued on next page*

*Figure 5 continued*

throughout the chromosome arm 2 R in type II neuroblasts in control or *fru*[C]-null brains. Fru peaks are shown along with Pc domain regions called on data using similar strategy as previously used to call canonical Pc domains (*Brown et al., 2018*). (**C**) Volcano plot showing fold-change of H3K27me3 signal in overall genomic regions in *fru*[C]-null brains versus control brains. (**D**) Volcano plot showing fold-change of H3K27me3 signal in regions bound by Fru[c] in *fru*[C]-null brains versus control brains. (**E**) Heatmaps are centered on Fru[c] summits with 2 kb flanking regions in genes uniquely transcribed in type II neuroblasts in control or *fru*[C]-null brains and ordered by signal intensity of H3K27me3 enrichment calculated from TMM-normalized tracks. (**F**) Left: Density plots showing proportion of all 500 bp regions in the genome not bound by Fru[c] covered by different amounts of H3K27me3 reads in *fru*[C]-null brains vs control brains. Right: Density plots showing proportion of all 500 bp Fru[c] bound regions covered by different amounts of H3K27me3 reads in *fru*[C]-null brains vs control brains. (**G**) Dotplot representing coverage of each Fru[c] peak not in Pc domains vs coverage of each Pc domain. The horizontal line in the volcano plot represents $-\log_{10}(0.05)=1.301$. All genes above this line have a FDR <0.05.

The online version of this article includes the following figure supplement(s) for figure 5:

**Figure supplement 1.** Levels of active histone marks are unchanged Fru[C]-bound regions.

by reducing *fru* function. Suppressor of zeste 12 (Su(z)12) and Chromatin assembly factor 1, p55 subunit (abbreviated as Caf-1) are two of the PRC2 core components. We performed CUT&RUN in control brains to determine whether regions enriched with Su(z)12 and Caf-1 overlap with Fru[C]-bound peaks. We found that Fru[C]-bound *cis*-regulatory elements in *Notch*, *dpn*, and *E(spl)mγ* display enrichment of Su(z)12 and Caf-1 (*Figure 6A*). Furthermore, most Fru[C]-bound peaks in genes uniquely transcribed in type II neuroblasts show Su(z)12 and Caf-1 enrichment (*Figure 6B*). Similarly, most Fru[C]-bound peaks in genes transcribed throughout the type II neuroblast lineage also shows Su(z)12 and Caf-1 enrichment (*Figure 6C*). These data support our model that Fru[C] functions through low levels of H3K27me3 enrichment to finely tune gene transcription in type II neuroblasts. Reducing PRC2 function alone does not lead to supernumerary neuroblast formation but strongly enhances the supernumerary neuroblast phenotype in *numb*[hypo] brains, identical to the results obtained by reducing *fru* function (*Figure 6D–G*; *Figure 4—figure supplement 1A*). Thus, reducing PRC2 activity increases activated Notch during asymmetric neuroblast division. These data led us to propose that Fru[C] functions together with PRC2 to finely tune gene expression in mitotic neuroblasts by promoting low-level enrichment of repressive histone marks in cis-regulatory elements.

## Discussion

Regulation of gene expression requires transcription factors and their associated chromatin-modifying activity, and a lack of insights into relevant transcription factors has precluded mechanistic investigations of gene expression by fine-tuning. By taking advantage of the well-established cell type hierarchy and sensitized genetic backgrounds in the type II neuroblast lineage, we demonstrated Fru[C] fine-tunes gene expression in neuroblasts. By focusing on the Notch signaling pathway in type II neuroblasts, we have been able to define generalizable mechanisms by which Fru[C] finely tunes gene expression levels using loss- and gain-of-function analyses. Our data indicate that Fru[C] likely functions together with PRC2 to dampen the expression of specific genes in mitotic neuroblasts by promoting low levels of H3K27me3 enrichment at their enhancers and promoters (*Figure 7*). We propose that local low-level enrichment of repressive histone marks can act to fine-tune gene expression.

### Fru is a multifaceted transcriptional regulator of gene expression

Fru protein isoforms have been detected in many cell types in flies (*Ito et al., 1996*; *Ryner et al., 1996*; *Djiane et al., 2013*; *Michki et al., 2021*; *Zhou et al., 2021*; *Dillon et al., 2022*; *Xu et al., 2022*). The role of Fru in stem cell differentiation remained undefined. Studies linking Fru isoform-specific DNA-binding across the genome with Fru function have been confounded by multiple issues. These include the co-expression of multiple isoforms with different binding specificities within the same cell types, as well as the heterogeneity and scarcity of cell types which express Fru within the central brain, where male-specific Fru isoforms (Fru[M]) have been most intensely studied (*Goodwin and Hobert, 2021*). In this study, the identification of Fru[C] as the sole isoform expressed in type II neuroblasts along with the use of *brat*-null brains, which are highly enrich for these neuroblasts, has enabled the study of Fru[C] genomic binding in a defined cell type where it is known to be expressed. By using gene activity in the Notch pathway as an in vivo functional readout, we have now been able to link genomic and genetic evidence to identify a clear role for Fru[C] in negatively regulating the

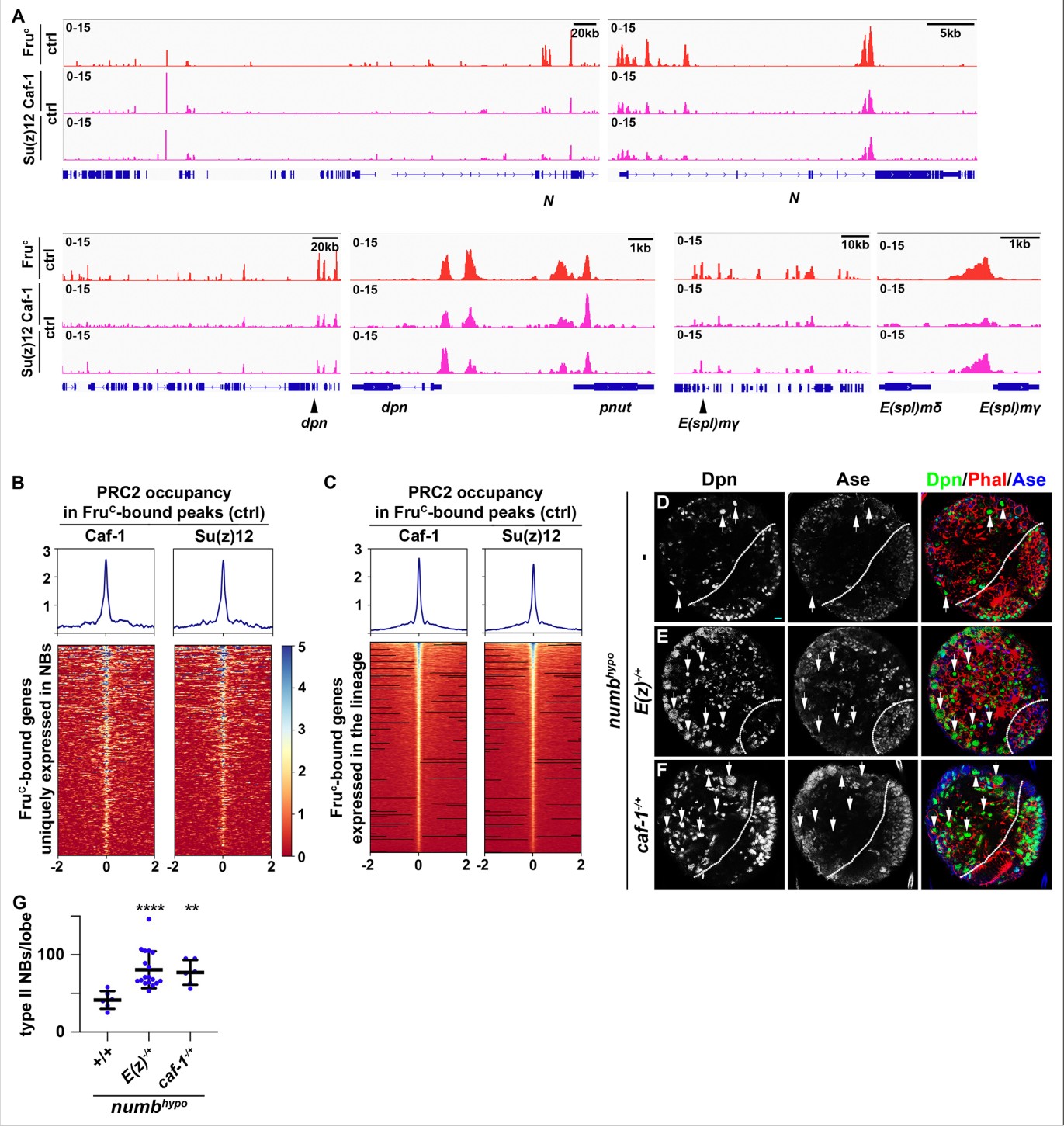

**Figure 6.** PRC2 subunits bind a high percentage of neuroblast-specific genes. (**A**) Representative z score-normalized genome browser tracks showing regions bound by Fru$^C$, Su(z)12, and Caf-1 in type II neuroblasts in control (*brat$^{11/Df(2L)Exel8040}$*) brains. Left: Zoomed-out images of the loci. Right: Zoomed-in images showing enrichment of PRC2 subunits in Fru$^C$-bound regions. (**B–C**) Heatmaps are centered on Fru$^c$ summits with 2 kb flanking regions in genes uniquely transcribed in type II neuroblasts in control brains and ordered by average signal intensity of Caf-1 and Su(z)12. Heatmap intensity is calculated from z score-normalized tracks. (**D–F**) The heterozygosity of E(z) (*E(z)$^{731/+}$*) or Caf-1 (*Caf-1$^{short/+}$*) enhances the supernumerary neuroblast phenotype in *numb$^{hypo}$* (*numb$^{ex112/15}$*) brains. (**G**) Quantification of total type II neuroblasts per brain lobe in *numb$^{hypo}$* (*numb$^{ex112/15}$*) brain alone or heterozygous for *E(z)* (*E(z)$^{731/+}$*) or *Caf-1* (*Caf-1$^{short/+}$*). Scale bars: 10 μm.

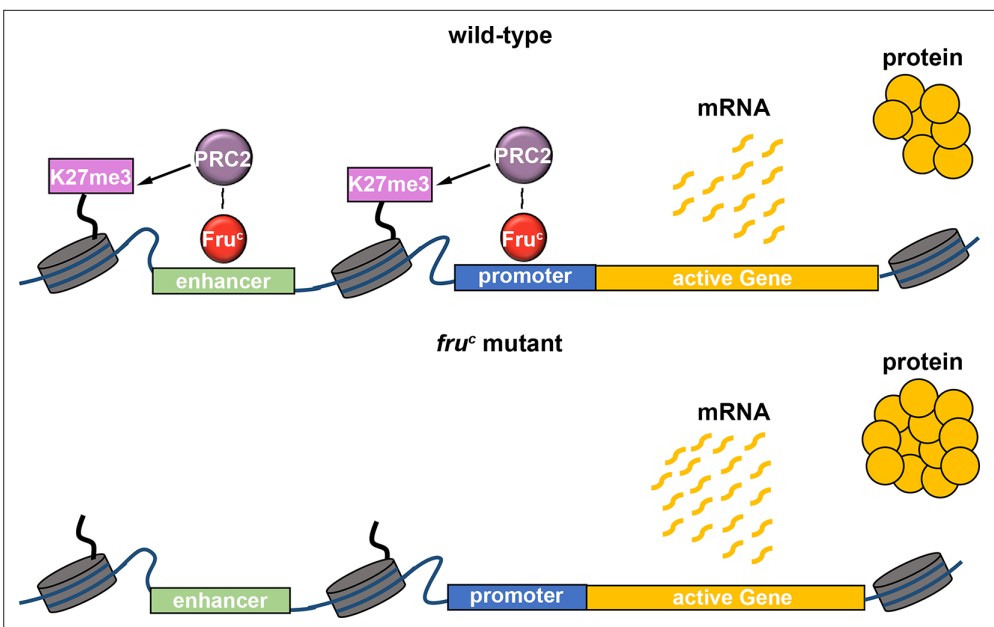

**Figure 7.** Fru[C] likely functions together with PRC2 to dampen the expression of stemness genes by promoting low levels of H3K27me3 at their *cis*-regulatory elements. Loss of *fru[c]* functions leads to reduced repressive histone marks and increased stemness gene expression in neuroblasts.

expression of Fru[C]-target genes during asymmetric neuroblast division (*Figures 3 and 4*). Our data strongly correlate the downregulation of *Notch* and Notch downstream-effector gene activity by Fru[C] to PRC2-mediated low-level enrichment of H3K27me3 (*Figures 5 and 6*). This mechanistic correlation appears to be broadly applicable to genes that promote stemness or prime differentiation in neuroblasts (*Figure 3D and E*). These data led us to propose a model in which Fru[C] functions together with PRC2 to fine-tune the expression of genes by promoting low-level enrichment of H3K27me3 at their *cis*-regulatory elements. Transcriptomic analyses have revealed that *fru* transcripts are enriched in fly renal stem cells in which Notch signaling plays an important role in regulating their stemness (*Xu et al., 2022*). We speculate that the mechanisms we have described in this study might be applicable to the regulation of gene expression in the renal stem cell lineage. It will also be interesting to investigate whether the male-specific Fru[MC] isoform, having the same DNA-binding specificity, utilizes a similar mechanism to regulate the multitude of developmental programs throughout the brain that contribute to a sexually dimorphic nervous system.

## Mechanisms that fine-tune gene transcription

What mechanisms allow transcriptional factors to promote inactivation of gene transcription vs. the fine-tuning of their expression? In the type II neuroblast lineage, transcription factors Erm and Hamlet (Ham) function together with histone deacetylases to sequentially inactivate type II neuroblast functionality genes including *tll* and *pointed* during INP commitment (*Weng et al., 2010*; *Zhu et al., 2011*; *Eroglu et al., 2014*; *Janssens et al., 2014*; *Koe et al., 2014*; *Xie et al., 2016*; *Hakes and Brand, 2020*; *Rives-Quinto et al., 2020*). In *erm*- or *ham*-null brains, Notch reactivation ectopically activates type II neuroblast functionality gene expression in INPs, driving their reversion into supernumerary neuroblasts. Importantly, mis-expressing either gene overrides activated Notch activity in neuroblasts, and drives them to prematurely differentiate into neurons. Thus, Erm- and Ham-associated histone deacetylation can robustly counteract activity of the Notch transcriptional activator complex and inactivate Notch downstream-effector gene transcription. In contrast to Erm and Ham, Fru[C] appears to reduce activity of the Notch transcriptional activator complex instead. Loss of *fru[c]* function modestly increases Notch activity in mitotic type II neuroblasts leading to moderately higher levels of Notch activity and Notch downstream-effector gene expression in immature INPs (*Figure 4A and E*). Ectopic Notch activity in immature INPs due to loss of *fru[c]* function can be efficiently buffered by the multilayered gene control mechanism and does not perturb the onset of INP commitment (*Figure 2F–I*).

Furthermore, neuroblasts continually overexpressing $Fru^C$ for 72 hours maintained their identities, despite displaying a reduced cell diameter and expressing markers that are typically diagnostic of $Ase^+$ immature INPs (*Figure 2J–L*). These results suggest that $Fru^C$ dampens rather than overrides activated Notch activity and are consistent with the findings that $Fru^C$-bound regions displaying little changes in histone acetylation levels between $fru^C$-null and control neuroblasts (*Figure 4—figure supplement 1C–D*). Thus, transcriptional repressors that inactivate gene transcription render the activity of transcriptional activators ineffective, whereas transcriptional repressors that fine-tune gene expression dampen their activity.

A key follow-up question on a proposed role for $Fru^C$ in fine-tuning gene expression in neuroblasts is the mechanistic link between this transcriptional repressor and the dampening of gene transcription. A previous study suggested that Fru functions through Heterochromatin protein 1a (Hp1a) to promote gene repression during the specification of sexually dimorphic neurons (*Ito et al., 2012*). Hp1a catalyzes deposition of the H3K9me3 mark (*Eissenberg and Elgin, 2014*). $Fru^C$-bound genes in neuroblasts display undetectable levels of H3K9me3, and knocking down *hp1a* function did not enhance the supernumerary neuroblast phenotype in *numb^hypo* brains (data not presented). Thus, it is unlikely that $Fru^C$ finely tunes gene transcription by promoting H3K9me3 enrichment. A small subset of FruC-bound peaks showed increased enrichment of H3K27ac in neuroblasts in $fru^C$-null brains compared with control brains (*Figure 4—figure supplement 1D*). However, $Fru^C$ overexpression mildly reduces activity of the Notch transcriptional activator complex in neuroblasts (*Figure 2J–L*). Thus, histone deacetylation appears to play a minor role in $Fru^C$-mediated fine-tuning of gene expression in neuroblasts. Most peaks that displayed statistically significantly reduced H3K27me3 levels in $fru^C$-null neuroblasts compared with control neuroblasts are bound by $Fru^C$ (*Figure 5E*). Many $Fru^C$-bound peaks displayed the enrichment of PRC2 subunits, Su(z)12 and Caf-1, and reduced PRC2 function enhanced the supernumerary neuroblast phenotype in *numb^hypo* brains (*Figure 6D–G*; *Figure 5—figure supplement 1A*). These results strongly suggest a model in which low-level enrichment of H3K27me3 in cis-regulatory elements of $Fru^C$-bound genes fine-tunes their expression in neuroblasts (*Figure 7*).

## PRC2 fine-tunes gene transcription during developmental transitions

A counterintuitive finding from this study is the role for PRC2 and low levels of H3K27me3 enrichment in fine-tuning active gene transcription in type II neuroblasts. PRC2 is thought to be the only complex that deposits the H3K27me3 repressive histone mark and functions to repress gene transcription (*Laugesen et al., 2019*; *Morgan and Shilatifard, 2020*; *Piunti and Shilatifard, 2021*). PRC2 subunits and H3K27me3 are enriched in many active genes in various cell types, including embryonic stem cells and quiescent B cells in mice and human differentiating erythroid cells (*Brookes et al., 2012*; *Frangini et al., 2013*; *Kaneko et al., 2013*; *Morey et al., 2013*; *Xu et al., 2015*; *Giner-Laguarda and Vidal, 2020*; *Ochiai et al., 2020*). However, the functional significance of their occupancies in active gene loci in vertebrate cells remains unclear due to a lack of sensitized functional readouts and a lack of insight regarding transcription factors for their recruitment. Several similarities exist between these vertebrate cell types and fly type II neuroblasts. First, both vertebrate and fly cells are poised to undergo a cell-state transition. Second, the pattern of PRC2 subunit occupancy in cis-regulatory elements of active vertebrate and fly genes appears as discrete peaks that are also enriched with low levels of H3K27me3. Building on the functional evidence collected in vivo, we propose that $Fru^C$ functions together with PRC2 to fine-tune the expression of genes that promote stemness or prime differentiation by promoting low-level enrichment of H3K27me3 in their *cis*-regulatory elements in type II neuroblasts (*Figure 7*). Thus, PRC2-mediated low-level enrichment of H3K27me3 we have described in this study should be broadly applicable to the fine-tuned gene activity attained by dampening transcription during binary cell fate specification and cell-state transitions in vertebrates.

An important question arising from our proposed model relates to how $Fru^C$ functions together with PRC2 to dampen gene transcription in neuroblasts. Studies in vertebrates have shown that loss of PRC2 activity mildly increases gene transcription levels, suggesting that PRC2 likely dampens gene transcription (*Morey et al., 2013*; *Pherson et al., 2017*). A separate study suggested a possible link between PRC2 and transcriptional bursts (*Ochiai et al., 2020*). The transcriptional activator complex binding to promoters affects burst sizes, whereas their binding to enhancers control burst frequencies (*Larsson et al., 2019*). The dwell time of the transcriptional activator complex bound to *cis*-regulatory elements directly affects the frequency, duration, and amplitude of transcriptional bursts. For example,

chromatin immunoprecipitation and live-cell imaging of fly embryos have suggested that the Su(H) occupancy time at *cis*-regulatory elements of Notch downstream-effector genes increases upon Notch activation (*Gomez-Lamarca et al., 2018*). Our genomic data indicate that Su(H)-bound regions in *Notch* and Notch downstream-effector genes that promote stemness in neuroblasts are bound by Fru[C] and are enriched for low levels of H3K27me3 and PRC2 subunits (*Figures 3F–G, 5A and 6A*). These observations suggest that Fru[C] and PRC2 might fine-tune expression of Notch pathway components by modulating transcriptional bursts in these loci. Examining levels of Su(H) enrichment at Fru[C]-bound peaks in Notch pathway component loci in *fru[C]*-null neuroblasts relative to control neuroblasts will enable the testing of this model.

# Materials and methods

**Key resources table**

| Reagent type (species) or resource | Designation | Source or reference | Identifiers | Additional information |
|---|---|---|---|---|
| Genetic reagent (*D. melanogaster*) | P{GawB}numb[NP2301] | Kyoto *Drosophila* Stock Center | 104153 | |
| Genetic reagent (*D. melanogaster*) | numb[ex115] | This study | | see "Fly genetics and transgenes" in Material and Methods |
| Genetic reagent (*D. melanogaster*) | fru[A]::Myc | *von Philipsborn et al., 2014* | | |
| Genetic reagent (*D. melanogaster*) | fru[B]::Myc | *von Philipsborn et al., 2014* | | |
| Genetic reagent (*D. melanogaster*) | fru[C]::Myc | *von Philipsborn et al., 2014* | | |
| Genetic reagent (*D. melanogaster*) | brat[11] | *Arama et al., 2000* | stock #: 97265 | |
| Genetic reagent (*D. melanogaster*) | brat[DG19310] | *Xiao et al., 2012* | | |
| Genetic reagent (*D. melanogaster*) | Df(2 L)Exel8040 | Bloomington *Drosophila* Stock Center | stock #: 7847 | |
| Genetic reagent (*D. melanogaster*) | numb[15] | *Berdnik et al., 2002* | | |
| Genetic reagent (*D. melanogaster*) | fru[sat15] | *Ito et al., 1996* | | |
| Genetic reagent (*D. melanogaster*) | fru[AJ96u3] | *Song et al., 2002* | | |
| Genetic reagent (*D. melanogaster*) | fru[ΔA] | *Neville et al., 2014* | | |
| Genetic reagent (*D. melanogaster*) | fru[ΔB] | *Neville et al., 2014* | | |
| Genetic reagent (*D. melanogaster*) | fru[ΔC] | *Billeter et al., 2006* | | |
| Genetic reagent (*D. melanogaster*) | E(z)[731] | *Anderson et al., 2011* | | |
| Genetic reagent (*D. melanogaster*) | Caf-1[short] | *Anderson et al., 2011* | | |
| Genetic reagent (*D. melanogaster*) | UAS-fru[C]::Myc | This study | | see "Fly genetics and transgenes" in Material and Methods |
| Genetic reagent (*D. melanogaster*) | UAS-fru[C]::ERD::Myc | This study | | see "Fly genetics and transgenes" in Material and Methods |

*Continued on next page*

*Continued*

| Reagent type (species) or resource | Designation | Source or reference | Identifiers | Additional information |
|---|---|---|---|---|
| Genetic reagent (*D. melanogaster*) | *UAS-dcr2; Wor-Gal4, Ase-Gal80; UAS-Stinger::RFP* | *Reichardt et al., 2018* | | |
| Genetic reagent (*D. melanogaster*) | *P{GawB}elavC155, P{UAS-mCD8::GFP.L}Ptp4ELL4, P{hsFLP}1* | Bloomington *Drosophila* Stock Center | stock #: 5146 | |
| Genetic reagent (*D. melanogaster*) | *P{GAL4-Act5C(FRT.CD2).P}S* | Bloomington *Drosophila* Stock Center | stock #: 4780 | |
| Genetic reagent (*D. melanogaster*) | *y[1] w[*]; P{ry[+t7.2]=neoFRT}82B P{w[+mC]=tubP-GAL80}LL3* | Bloomington *Drosophila* Stock Center | stock #: 5135 | |
| Genetic reagent (*D. melanogaster*) | *w[1118]; P{y[+t7.7] w[+mC]=GMR9D11-GAL4}attP2* | Bloomington *Drosophila* Stock Center | stock #: 40731 | |
| Genetic reagent (*D. melanogaster*) | *UAS-E(spl)mγ*$_{RNAi}$ | Bloomington *Drosophila* Stock Center | stock #: 25978 | |
| Antibody | anti-GFP (chicken polyclonal) | Aves Labs | Cat# GFP-1010 RRID:AB_2307313 | IF (1:2000) |
| Antibody | anti-Ase (rabbit polyclonal) | *Weng et al., 2010* | | IF (1:400) |
| Antibody | anti-Fru$^{COM}$ (rabbit polyclonal) | *Ito et al., 2012* | | IF (1:500) CUT-&-RUN (1 µl) |
| Antibody | anti-Trl (rabbit polyclonal) | *Judd et al., 2021* | | IF (1:500) CUT-&-RUN (1 µl) |
| Antibody | anti-cMyc (mouse polyclonal) | Sigma | SKU: M4439 | IF (1:200) |
| Antibody | anti-Su(H) (mouse polyclonal) | Santa Cruz | SKU: 398453 | IF (1:100), |
| Antibody | anti-Pros (mouse monoclonal) | *Lee et al., 2006a* | | IF (1:500) |
| Antibody | anti-dpn (rat monoclonal, ascite) | *Weng et al., 2010* | | IF (1:1000) |
| Antibody | anti-Mira (rat monoclonal) | *Lee et al., 2006a* | | IF (1:100) |
| Antibody | anti-cMyc (goat polyclonal) | Abcam | Cat#: ab9132 | CUT-&-RUN (1 µl) |
| Antibody | anti-Caf-1 (rabbit polyclonal) | *Tyler et al., 1996* | | CUT-&-RUN (1 µl) |
| Antibody | anti-Su(z)12 (rabbit polyclonal) | *Loubière et al., 2016* | | CUT-&-RUN (1 µl) |
| Antibody | anti-IgG (rabbit polyclonal) | EpiCypher | | CUT-&-RUN (1 µl) |
| Antibody | anti-H3K9me3 (rabbit polyclonal) | Abcam | Cat#: ab8898 | CUT-&-RUN (1 µl) |
| Antibody | anti-H3K4me3 (rabbit polyclonal) | Active Motif | Cat#: 39159 | CUT-&-RUN (1 µl) |
| Antibody | anti-H3K27me3 (rabbit polyclonal) | Sigma Aldrich | Cat#: 07–449 | CUT-&-RUN (1 µl) |
| Antibody | anti-H3K27ac (rabbit polyclonal) | Active Motif | Cat#: 39136 | CUT-&-RUN (1 µl) |
| Chemical compound | Rhodamine Phalloidin | Invitrogen | Cat#: R415 | IF (1:100) |
| Chemical compound | Alexa Fluor Plus 405 Phalloidin | Invitrogen | Cat#: A30104 | |
| Chemical compound | DRAQ5 | Abcam | Cat#: ab108410 | |
| Chemical compound | Papain | Millipore Sigma | Cat#: P4762-25MG | |
| Chemical compound | Collagenase type I | Millipore Sigma | Cat#: SCR103 | |
| Chemical compound | E-64 | Millipore Sigma | Cat#: E3132-1MG | |
| Chemical compound | Fetal Bovine Serum | Millipore Sigma | Cat#: F0926-50ML | |
| Chemical compound | Schneider's Media | Millipore Sigma | Cat#: S0146-500ML | |
| Chemical compound | ProLong Gold Antifade Mountant | Invitrogen | Cat#: P36930 | |

*Continued*

| Reagent type (species) or resource | Designation | Source or reference | Identifiers | Additional information |
|---|---|---|---|---|
| Chemical compound | ProLong Gold Antifade Mountant with DNA Stain DAPI | Invitrogen | Cat#: P36935 | |
| Chemical compound | Agencourt AMPure XP - 5 mL | Beckman Coulter | Cat#: A63880 | |
| Chemical compound | Sodium butyrate | Sigma Aldrich | Cat#: B5887 | |
| Commercial assay or kit | 10x chromium v3 single-cell gene expression kit | 10x Genomics | Cat# 1000154 | |
| Commercial assay or kit | B1-labeled HCR RNA-FISH intron probe (*Notch*) | Molecular Instruments, Inc | dm6, chrX: 3159233–3163479 | |
| Commercial assay or kit | B1-labeled HCR RNA-FISH intron probe (*dpn*) | Molecular Instruments, Inc | dm6, chr2R: 8230502–8231599 & 8231694–8231923 | |
| Commercial assay or kit | B1-labeled HCR RNA-FISH intron probe (*klu*) | Molecular Instruments, Inc | dm6, chr3L: 10985072–10996227 | |
| Commercial assay or kit | B1-labeled HCR RNA-FISH intron probe (*CycE*) | Molecular Instruments, Inc | dm6, chr2L: 15731785–15737628 | |
| Commercial assay or kit | HCR RNA-FISH amplifiers | Molecular Instruments, Inc | | |
| Commercial assay or kit | CUTANA ChIC/CUT&RUN Kit | Epicypher | SKU: 14–1048 | |
| Commercial assay or kit | NEBNext Ultra II DNA Library Prep Kit for Illumina | New England Biolabs | Cat#: E7645L | |
| Commercial assay or kit | NEBNext Multiplex Oligos for Illumina (96 Unique Dual Index Primer Pairs) | New England Biolabs | Cat#: E6440L | |
| Software, algorithm | Script Files | This Study | GITHUB | |
| Software, algorithm | Cell Ranger (6.0.1) | 10x Genomics | RRID:SCR_017344 | |
| Software, algorithm | Fiji/ImageJ | *Schindelin et al., 2012* | RRID:SCR_002285 https://fiji.sc/ | |
| Software, algorithm | scanpy scRNA-seq analysis software | *Wolf et al., 2018* | RRID:SCR_018139 | |
| Software, algorithm | Harmony | *Korsunsky et al., 2019* | | |
| Software, algorithm | Bioinfokit | Renesh Bedre, 2020, March 5 | | |
| Software, algorithm | Matplotlib | https://zenodo.org/badge/ https://doi.org/10.5281/zenodo.592536.svg | RRID:SCR_008624 | |
| Software, algorithm | Seaborn | https://doi.org/10.21105/joss.03021 | RRID:SCR_018132 | |
| Software, algorithm | FastQC | *Wingett and Andrews, 2018* | RRID:SCR_014583 | |
| Software, algorithm | cutadapt | *Martin, 2011* | RRID: SCR_011841 | |
| Software, algorithm | Samtools | *Li et al., 2009* | RRID: SCR_002105 | |
| Software, algorithm | MACS2 | *Zhang et al., 2008* | RRID: SCR_013291 | |
| Software, algorithm | CUT-RUNTools-2.0 | *Yu et al., 2021* | | |
| Software, algorithm | bedtools | *Quinlan and Hall, 2010* | RRID: SCR_008848 | |
| Software, algorithm | HOMER | *Heinz et al., 2010* | RRID: SCR_010881 | |
| Software, algorithm | deeptools | *Ramírez et al., 2016* | RRID: SCR_016366 | |
| Software, algorithm | Integrative Genomics Viewer | *Robinson et al., 2011* | RRID: SCR_011793 | |
| Software, algorithm | GoPeaks | *Yashar et al., 2022* | | |

*Continued on next page*

*Continued*

| Reagent type (species) or resource | Designation | Source or reference | Identifiers | Additional information |
|---|---|---|---|---|
| Software, algorithm | SICER | *Zang et al., 2009* | RRID:SCR_010843 | |
| Software, algorithm | featureCounts | *Liao et al., 2014* | RRID: SCR_012919 | |
| Software, algorithm | EdgeR | *Robinson et al., 2010* | RRID: SCR_012802 | |
| Software, algorithm | diffReps | *Shen et al., 2013* | RRID:SCR_010873 | |
| Software, algorithm | iCisTarget | *Imrichová et al., 2015* | | |
| Software, algorithm | XSTREME | *Grant and Bailey, 2021* | RRID:SCR_001783 | |
| Software, algorithm | LAS AF | Leica Microsystems | RRID:SCR_013673 | |

## Fly genetics and transgenes

Fly crosses were carried out in 6-oz plastic bottles at 25 °C, and eggs were collected in apple caps in 8 hr intervals (4 hr for scRNA-seq). Newly hatched larvae were genotyped by presence or absence of the balancer chromosome *CyO,Act-GFP* and/or *TM6B,Tb* and cultured on caps containing corn-meal *Drosophila* culturing media for 96 hr. For GAL4 based overexpression or knock down experiments, larvae were shifted to 33 °C after eclosion to induce transgene expression via suppression of *tub-Gal80^{ts}*.

Larvae for MARCM analyses (*Lee and Luo, 2001*) were genotyped after eclosion and allowed to grow at 25 °C for 24 hr. Corn meal caps containing larvae were then placed in a 37 °C water bath for 90 min to induce clones. Heat-shocked larvae were allowed to recover and grow at 25 °C for 72 hr prior to dissection.

For CUT&RUN experiments, fly crosses were carried out in 30oz fly condos, and eggs were collected on 10 mm apple caps in 12 hr intervals. Newly hatched larvae were genotyped and cultured on corn meal caps for ~5–6 days.

The following transgenic lines were generated in this study: *UAS-fru^C::Myc* and *UAS-ERD::fruC^{zf}::Myc*. The DNA fragments were cloned into *p{UAST}attB*. The transgenic fly lines were generated via *φC31* integrase-mediated transgenesis (*Bischof and Basler, 2008*). *numb^{EX112}* alleles were generated by imprecise excision of *P{GawB}numb^{NP2301}*, which was inserted at a P-element juxtaposed to the transcription start site of the *numb* gene. The excised regions were determined by PCR followed by sequencing.

## Immunofluorescent staining and antibodies

Larvae brains were dissected in phosphate buffered saline (PBS) and fixed in 100 mM PIPES (pH 6.9), 1 mM EGTA, 0.3% Triton X-100 and 1 mM $MgSO_4$ containing 4% formaldehyde for 23 min. Fixed brain samples were washed with PBST containing PBS and 0.3% Triton X-100. After removing fixation solution, samples were incubated with primary antibodies for 3 hrs at room temperature. After 3 hr, samples were washed with PBST and then incubated with secondary antibodies overnight at 4 °C. The next day, samples were washed with PBST and equilibrated in ProLong Gold antifade mount (Thermo Fisher Scientific). Antibodies used in this study include chicken anti-GFP (1:2000; Aves Labs, SKU 1020), rabbit anti-Ase (1:400) (*Weng et al., 2010*), rabbit anti-Fru^{COM} (1:500; D. Yamamoto), rabbit anti-Trl (1:500; J.T. Lis), mouse anti-cMyc (1:200; Sigma, SKU: M4439), mouse anti-Su(H) (1:100; Santa Cruz, SKU: 398453), mouse anti-Pros (1:500) (*Lee et al., 2006c*), rat anti-Dpn (1:1000) (*Weng et al., 2010*), and rat anti-Mira (1:100) (*Lee et al., 2006c*). Secondary antibodies were from Jackson ImmunoResearch Inc and Thermo Fisher Scientific. We used rhodamine phalloidin or Alexa Fluor Plus 405 phallloidin (ThermoFisher Scientific) to visualize cortical actin. Confocal images were acquired on a Leica SP5 scanning confocal microscope (Leica Microsystems Inc) using a 63 x glycerol immersion objective, as z-stacks with 1.51 μm thickness. Images were taken at 1.5 x zoom for whole lobe, or 3/5 x zoom for single neuroblasts/single clonal lineages.

## Hybridization chain reaction (HCR) and immunofluorescent staining

mRNA signals in the larval brain were developed by performing in situ HCR v3.087. We modified the protocol of in situ HCR v3.0 to combine immunofluorescent staining of the larval brain. Third instar

larval brains were dissected in PBS and fixed in 100 mM PIPES (pH 6.9), 1 mM EGTA, 0.3% Triton X-100 and 1 mM $MgSO_4$ containing 4% formaldehyde for 23 min. Fixed brain samples were washed with PBST containing PBS and 0.3% Triton X-100. After removing fix solution, samples were pre-hybridized with hybridization buffer (10% formamide, 5×SSC, 0.3% Triton X-100 and 10% dextran sulfate) at 37 °C for 1 hr. Prehybridized samples were mixed with 5 nM Sp-1 mRNA HCR probe (Molecular Instruments, Los Angeles, CA) and incubated at 37 °C overnight. After hybridization, samples were washed with washing buffer (10% formamide, 5×SSC, 0.3% Triton X-100) and then incubated with amplification buffer (5×SSC, 0.3% Triton X-100 and 10% dextran sulfate) at 25 °C for 30 min. During washing period, imager hairpins (Molecular Instruments, Los Angeles, CA) were denatured at 95 °C for 2 mins. Once samples were equilibrated in amplification buffer, samples were mixed with 3 μM of denatured imager hairpins and incubated at 25 °C for overnight. The next day, samples were washed with PBST and then re fixed in 100 mM PIPES (pH = 6.9), 1 mM EGTA, 0.3% Triton X-100 and 1 mM $MgSO_4$ containing 4% formaldehyde for 15 min to initiate immunofluorescent staining procedures.

## Quantification and statistical analyses

All biological replicates were independently collected and processed. The observers were blind to the genotypes. All brain samples, except those damaged during processing, will be included in data analyses. Only one brain lobe per brain was imaged to ensure measurement biological variability. Cell types were counted by the imager taker based on the presence of expected markers and cell size (NB:>7 μm diameter; type II NB: Dpn$^+$,Ase$^-$; GMC: Pros$^+$,Ase$^+$) for cells that were outside of the optic lobe region (identified by morphology). All statistical analyses were performed using a two-tailed Student's t-test, and p-values <0.05, <0.005, <0.0005, and <0.00005 are indicated by (*), (**), (***) and (****), respectively in figures. GraphPad Prism was used to generate dot-plots.

For newborn immature INP identification, Dpn protein was used to identify the type II neuroblast and Mira was used to identify the newly born immature INP nucleus. Pixel intensities of the proteins of interest were measured in the nucleus of the type II neuroblast and corresponding newly born immature INP using ImageJ software. The relative pixel intensity of the protein of interest in the immature INP was taken in relation to the pixel intensity in the type II neuroblast. All biological replicates were independently collected and processed.

For smFISH data quantification, brain lobes were stained for DAPI using ProLong Gold antifade mount with DAPI (ThermoFisher Scientific). The top 8 most dorsal neuroblasts were selected from individual brain lobes from distinct brains by cell size and morphology. Fluorescent smFISH foci which overlapped with the nuclear DAPI stain were counted if they were larger than 3 μm in diameter in the z-axis direction (persist for more than 2 slices).

## scRNA-seq of the type II neuroblast lineage

*UAS-dcr2; Wor-gal4, Ase-gal80; UAS-RFP::stinger* larval brains (n=50) were dissected 96 hr after larval hatching in ice cold Rinaldini's solution during a 45 min interval. Dissected brains were transferred to Eppendorf tubes containing 30 μL of Rinaldini's solution. A total of 10 μL of 20 mg/mL papain, 10 μL of 20 mg/mL type-1 collagenase, and 1 μL of 15 μM ZnCl was added to the tube. Additional Rinaldini's solution was added to adjust the final volume to be 100 μL. The tube was mixed gently by flicking, and them incubated on a heat block at 37 °C for 1 hr, while covered with aluminum foil. During this incubation, the tube was flicked for mixing every 10 min.

After the 1 hr incubation, 5 μL of 100 μM E-64 was added to stop the papain digestion. Samples were incubated on ice for 2 min, and then centrifuged for 3 min at 500 *g*. Supernatant was carefully removed, and chemical dissociated brains were resuspended in 100 μL Schneider's media with 10% fetal bovine serum (FBS). Mechanical dissociation was performed by setting a P100 pipette to 70 μL and titrating 30 times at a frequency of ~1 Hz. After titration, cells were diluted with 400 μL Schneider's media with 10% FBS, bringing the total volume to 500 μL. A total of 1 μL of DRAQ5 DNA stain (Thermo Fisher Scientific) was added to label cells apart from debris.

A Sony MA900 FACS machine was used to select for RFP+, DRAQ5 + cells. Cells were sorted into a 1.5 mL Eppendorf tube prefilled with 100 μL of Schneider's media with 10% FBS. Approximately 30,000 RFP$^+$, DRAQ5 events were sorted. Cells were transported on ice to the University of Michigan's Advanced Genomics Core and were loaded for 10x Chromium V3 sequencing following the manufacturer's instructions.

The mRNA was subsequently reverse-transcribed, amplified, and sequenced on an Illumina-NovaSeq-6000 chip (University of Michigan Advanced Genomics Core). Then, 151 bp paired-end sequencing was performed, with a target of 100,000 reads/cell.

## Data analysis for scRNA-seq

Reads were mapped using Cell Ranger (6.0.1) to the *Drosophila* genome assembly provided by ENSEMBL, build BDGP6.32, with DsRed (Genbank: AY490568) added to the genome.

Downstream scRNA-seq analyses was performed using SCANPY (*Wolf et al., 2018*). Count matrices were concatenated between our dataset and previously published previously published scRNA-seq data generated from INPs and downstream progeny of the type II lineage (*Michki et al., 2021*). The top 2,000 highly variable genes were identified, and then principal component analysis was performed using these highly variable genes with 50 components. The dataset was then harmonized using Harmony (*Korsunsky et al., 2019*), and completed in four iterations. Neighborhood identification was computed with k=20, and then UMAP (*Becht et al., 2018*) was performed using a spectral embedding of the graph. Finally, clusters were identified using the Leiden algorithm (*Traag et al., 2019*), with a resolution of 1. Cell-type annotation was performed by further clustering of clusters 1 and 14, and labeling was performed based on known marker genes.

Pseudotime analysis was performed by calculating the diffusion pseudotime (*Haghverdi et al., 2016*) trajectory implementation in SCANPY, using an initial root cell selected as $dpn^+$, $pnt^+$, $DsRed^+$ and visually based on UMAP (ID: ‘CATTCTAAGCAACTTC-1-0’).

Differential expression analysis for genes between neuroblasts and immature INPs was completed by separating cluster 14 using Leiden with a resolution of 0.3. SCANPY's `rank_gene_groups` was used with `method='t-test-overestim_var'` to determine log-fold changes and adjusted p-values of expressed genes. Differentially expressed genes were defined as having |Fold Change|>2 and adjusted -value <0.05, and invariant genes were defined as having |Fold Change|<2. Bioinfokit (Renesh Bedre, 2020, March 5) was used to generate the volcano plot figure.

## CUT&RUN on neuroblast-enriched brains

$brat^{11/Df}$; $fru^C$::Myc (control) or $brat^{11/Df}$; $fru^{ΔC/Aj96u3}$ ($fru^{ΔC/-}$) larval brains were dissected in 45 min time windows in PBS and transferred to 0.5 mL Eppendorf tubes. Dissected brains were then collected at the bottom of the tube, and supernatant was removed. CUTANA ChIC/CUT&RUN (Epicypher) was performed per the manufacturer's protocol, with modifications. Brains were resuspended in 100 µL wash buffer, and then homogenized by ~30–50 passes of a Dounce homogenizer. Homogenized samples were transferred to a 1.5 mL Eppendorf tube, and then the cells were pelleted by centrifugation (600 g for 3 min). Next, then pellets were processed using the CUT&RUN kit. A total of 0.5 ng of antibody was used per sample, (or 0.5 µL if antibody concentration was unknown). A total of 0.5 ng of *Escherichia coli* spike-in DNA was added into each sample as a spike-in control.

For control brains, antibodies used were goat anti-cMyc (Abcam, ab9132), mouse anti-Su(H), rabbit anti-Trl, rabbit anti-Caf-1 (Gift from J. Kadonaga, *Tyler et al., 1996*), rabbit anti-Su(z)12 (Gift from G. Cavalli, *Loubière et al., 2016*), rabbit anti-IgG (Epicypher), and rabbit anti-H3K9me3 (Abcam, ab8898). For both control and $fru^{-/-}$ brains, antibodies used were rabbit anti-H3K4me3 (Active Motif: 39159), rabbit anti-H3K27me3 (Sigma Aldrich, 07–449), and rabbit anti-H3K27ac (Active Motif: 39136). A total of 50 brains were collected for transcription factor samples and 25 brains were collected for histone mark samples. All samples were performed in duplicate. Samples targeting acetylation had 100 mM of sodium butyrate (Sigma Aldrich) added to all buffers. Samples using mouse antibodies underwent an additional antibody incubation step, where samples were washed 2 x with cell permeabilization buffer after primary antibody incubation, and then were incubated for 1 hr with 0.5 ng of rabbit anti-mouse IgG (Abcam, ab46540).

Fragmented DNA was diluted to 50 µL in 0.1 x TE and library prepped using the NEBNext Ultra II DNA Library Prep Kit for Illumina (E7645) with NEBNext Multiplex Oligos for Illumina (E6440) per the manufacturer's protocol with modifications. The adaptor was diluted 1:25, and bead clean-up steps were performed using 1.1 x AMPure Beads without size selection. The PCR cycle was modified to match specifications provided by the CUTANA ChIC/CUT&RUN kit. DNA was eluted in 20 µL 0.1 x TE.

DNA samples were assessed for concentration and quality on an Agilent TapeStation. Samples with greater than 1% adaptor underwent an additional round of bead cleanup. Samples that passed

quality control were sequenced on an IlluminaNovaSeq-6000 chip (University of Michigan Advanced Genomics Core). Then, 151 bp paired end sequencing was performed, with a target of at least 10,000,000 reads per replicate.

## CUT&RUN data analysis

Read quality was checked using FastQC (*Wingett and Andrews, 2018*). Reads were trimmed using cutadapt (*Martin, 2011*), and aligned to BDGP6.32 using bowtie2 (*Langmead and Salzberg, 2012*) with the flags `-local, --very-sensitive, --no-mixed, --no-discordant -dovetail`, and `-I 10 -X 1000`. Samtools (*Li et al., 2009*) was used to convert file formats and to mark fragments less than 120 bp.

For transcription factor samples, only reads with a fragment size <120 bp were kept for downstream analysis. Peaks were called individually on each replicate by MACS version 2 (*Zhang et al., 2008*), using parameters specified in CUT&RUNTools2 (*Yu et al., 2021*), and then merged using bedtools (*Quinlan and Hall, 2010*). Downstream analyses were carried out with both Fru$^c$::Myc and Fru$^{COM}$ peak sets and showed similar results. The data shown in heatmaps use the Fru$^c$::Myc peakset. For Fru$^c$ peaks, only high confidence peaks (-$\log_{10}$(Q-value) >100) were kept for downstream analysis. MACS2 was similarly used to call peaks on H3K9me3 samples, and high confidence peaks (-$\log_{10}$(Q-value) >100) were merged using bedtools `merge -d 3000`. We then blacklisted our transcription factor peak sets against H3K9me3 peaks to remove heterochromatin regions from the downstream analysis. High signal and low mappability regions defined by the ENCODE Blacklist (*Amemiya et al., 2019*) were also removed using bedtools. A random peak set was generated by calling bedtools `shuffle` on the Fru$^c$::Myc peakset, to generate a background control that covered the same number of regions and same number of bp's as Fru$^c$::Myc peakset but at new randomly determined genomic loci. An additional random peak set was generated using the same method, but starting with Su(H) peaks.

Peak sets were annotated using HOMER (*Heinz et al., 2010*), which was used to determine the genomic distribution of the transcription factors and genes associated with each peak. Regulatory regions were defined as peaks in either intronic or intergenic regions. Fru$^c$ -bound genes were determined as genes that had a Fru$^c$::Myc peak annotated as being associated to that gene. These peaks were also classified as immature INP-enriched, invariant, or neuroblast enriched peaks based on their corresponding gene's classification from our single-cell data. This same process was repeated for the randomized peak sets.

Bigwig files for transcription factors were generated using deeptools (*Ramírez et al., 2016*) `bamCoverage` with flags `--ignoreDuplicates --maxFragmentLength 120 --normalizeUsing RPKM`. Correlation between Fru$^c$::Myc and Fru$^{COM}$ was calculated using deeptools `mutiBigWigSummary` with default parameters. Correlation was plotted on log-log axes using deeptools `plotCorrelation` with the flag `--log1p` and otherwise default parameters. Bigwig tracks were visualized using the Integrative Genomics Viewer (IGV) (*Robinson et al., 2011*). For data visualization, z score-normalized bigwigs were generated by subtracting the mean read coverage (counts) from the merged replicate read counts in 10 bp bins across the entire genome and dividing by the standard deviation (*Larson et al., 2021*). Heatmaps for Fru$^c$::Myc signal at open chromatin regions, Su(H) peaks, and Trl peaks were generated using deeptools. Heatmaps for Su(z)12 and Caf-1 were generated at Fru$^c$ peaks which are associated with neuroblast enriched genes and for all Fru$^c$ peaks.

TMM (Trimmed Mean of M-Values) normalization was performed on histone mark samples to accurately account for differences in library composition and sequencing depth. First, peaks were called on control samples against IgG using GoPeaks (*Yashar et al., 2022*). featureCounts (*Liao et al., 2014*) was then used to count reads from each replicate inside the peaks for each histone mark across control and *fru$^{-/-}$* samples. EdgeR (*Robinson et al., 2010*) `calcNormFactors (method = TMM)` was called on each histone mark count matrix, and the final normalization factor was calculated as 1,000,000/ (normFactor * number of reads in peaks). Bigwig files were generated for individual replicates by using deeptools bamcoverage on aligned bam files with the `--scaleFactor` equal to the final normalization factor that was calculated. Bigwigs were then merged using deeptools `bigWigCompare`, and z score bigwig files for histone marks were generated and visualized in IGV. Heatmaps for H3K27me3 between control (*brat$^{-/-}$*) and (*brat$^{-/-}$; fru$^{ΔC/-}$*) were generated at Fru$^c$ peaks which are associated with neuroblast enriched genes and for all Fru$^c$ peaks.

Differential enrichment for each histone marks between control ($brat^{-/-}$) and ($brat^{-/-}$; $fru^{\Delta C/-}$) was performed using diffReps (*Shen et al., 2013*) with a 500 bp sliding window and otherwise default parameters. Bins which overlapped with $Fru^c$::Myc peaks were marked as $Fru^c$ bound. Volcano plots showing $-Log_{10}$(pval-adj) vs $Log_2$FoldChange were visualized using bioinfokit (Renesh Bedre, 2020).

Bash code was used to generate a bed file which covered the genome in 500 bP bins. These bins were intersected with bedtools to determine bins that were bound or unbound by $Fru^c$::Myc peaks. H3K27me3 bam files from control and $fru^{\Delta C/-}$ were subset using bash and samtools to contain 10 million fragments each. Reads in Fru bound or unbound bins were then counted using bedtools `multicov` on the subset bam files. Density plots of reads / bin were visualized in Python using matplotlib and seaborn (https://doi.org/10.21105/joss.03021).

For H3K27me3 control data, canonical Polycomb domains were called using SICER (*Zang et al., 2009*) with parameters `-w 500 f 0 -egf 0.7 g 2000` based on a previous study of Polycomb domains (*Brown et al., 2018*). Peaks that had a score higher than 500 from SICER, and were larger than 3 k-bp were kept, and merged across peaks within 10 k-bp and between replicates. Bedtools was used to determine $Fru^c$::Myc peaks outside of Polycomb domains. Finalized regions were quantified by counting number of reads from H3K27me3 control brains in respective regions using feature-Counts. Afterwards, reads in each region were normalized by region length, and average number of reads was calculated across replicates.

## Motif analyses

Motifs were searched for within +/-100 b p of $Fru^{COM}$ for *Drosophila* position weight matricies (PWMs) using iCisTarget (*Imrichová et al., 2015*) with default parameters. Analyses were run using all $Fru^{COM}$ peaks, or only $Fru^{COM}$ peaks associated with promoters or regulatory regions, and the top motif was selected for further analysis. The top motif found using all peaks and using regulatory peaks was the same. The normalized enrichment score NES, [AUC-μ]/σ was recorded for the motifs. Motif PWMs were obtained and motif locations in the genome were calculated using HOMER scanMotifGenome-Wide with a log odd detection threshold of 8. $Fru^{COM}$ and random peaks were extended by +/-200 bp and the percent of peaks containing motifs was calculated. De novo motif searching was attempted using XSTREME (*Grant and Bailey, 2021*) and HOMER, but no motifs were confidently identified.

## Acknowledgements

We thank Drs. G Cavalli, J Kadonaga, J Lis, and D Yamamoto for providing us with reagents. We thank the Advance Genomics Core and the Flow Cytometry core for technical assistance. We thank Drs. N Michki and Y Li for technical advice and assistance on the single-cell RNA sequencing experiment. We thank Drs. Chris Q Doe and Derek H Janssen and members of the Lee lab for helpful discussions, and Science Editors Network for editing the manuscript. We thank the Bloomington *Drosophila* Stock Center, Kyoto Stock Center, and the Vienna *Drosophila* RNAi Center for fly stocks. We thank Best-Gene Inc and GenetiVision Corp. for generating the transgenic fly lines. This work is supported by NIH grants R01NS107496 and R01NS111647. SFG and MN are supported by a Wellcome Trust Senior Investigator Award (106189/Z/14/Z) to SFG.

## Additional information

### Funding

| Funder | Grant reference number | Author |
| --- | --- | --- |
| National Institute of Neurological Disorders and Stroke | R01NS107496 | Cheng-Yu Lee |
| National Institute of Neurological Disorders and Stroke | R01NS111647 | Melissa M Harrison |
| Wellcome Trust | 106189/Z/14/Z | Stephen F Goodwin |

| Funder | Grant reference number | Author |
|--------|------------------------|--------|

The funders had no role in study design, data collection and interpretation, or the decision to submit the work for publication. For the purpose of Open Access, the authors have applied a CC BY public copyright license to any Author Accepted Manuscript version arising from this submission.

## Author contributions

Arjun Rajan, Conceptualization, Data curation, Software, Formal analysis, Validation, Investigation, Visualization, Methodology, Writing – original draft, Project administration, Writing – review and editing; Lucas Anhezini, Conceptualization, Data curation, Formal analysis, Validation, Investigation, Visualization, Methodology, Project administration; Noemi Rives-Quinto, Conceptualization, Data curation, Formal analysis, Validation, Investigation, Visualization, Project administration; Jay Y Chhabra, Investigation; Megan C Neville, Resources, Methodology, Writing – original draft; Elizabeth D Larson, Resources, Methodology; Stephen F Goodwin, Resources, Funding acquisition, Writing – original draft; Melissa M Harrison, Resources, Funding acquisition, Methodology, Writing – original draft; Cheng-Yu Lee, Conceptualization, Data curation, Supervision, Funding acquisition, Writing – original draft, Project administration, Writing – review and editing

## Author ORCIDs

Arjun Rajan ⓘ http://orcid.org/0000-0001-7043-1031
Megan C Neville ⓘ http://orcid.org/0000-0001-8506-9944
Stephen F Goodwin ⓘ http://orcid.org/0000-0002-0552-4140
Melissa M Harrison ⓘ http://orcid.org/0000-0002-8228-6836
Cheng-Yu Lee ⓘ http://orcid.org/0000-0003-2291-1297

## Decision letter and Author response

Decision letter https://doi.org/10.7554/eLife.86127.sa1
Author response https://doi.org/10.7554/eLife.86127.sa2

## Additional files

### Supplementary files

• Supplementary file 1. Raw data and statistics for all quantified data.

• Supplementary file 2. Differential gene expression values for Neuroblasts vs immature INPs from scRNA-seq.

• MDAR checklist

### Data availability

Sequencing data have been deposited in GEO under accession codes GSE218257. All quantifications are provided in Supplementary File 1. All analysis code used has been deposited in GitHub (copy archived at *Rajan et al., 2023*).

The following dataset was generated:

| Author(s) | Year | Dataset title | Dataset URL | Database and Identifier |
|-----------|------|---------------|-------------|-------------------------|
| Rajan A, Anhezini L, Rives-Quinto N, Chhabra JY, Neville MC, Larson ED, Goodwin SF, Harrison MM, Lee CY | 2023 | Low-level repressive histone marks fine-tune stemness gene transcription in neural stem cells | http://www.ncbi.nlm.nih.gov/geo/query/acc.cgi?acc=GSE218257 | NCBI Gene Expression Omnibus, GSE218257 |

The following previously published datasets were used:

| Author(s) | Year | Dataset title | Dataset URL | Database and Identifier |
|---|---|---|---|---|
| Michki NS, Li Y, Sanjasaz K, Zhao Y, Shen F, Walker LA, Cao W, Lee C, Cai D | 2021 | The molecular landscape of neural differentiation in the developing *Drosophila* brain revealed by targeted scRNA-seq and multi-informatic analysis | http://www.ncbi.nlm.nih.gov/geo/query/acc.cgi?acc=GSE153723 | NCBI Gene Expression Omnibus, GSE153723 |
| Larson ED, Komori H, Gibson TJ, Ostgaard CM, Hamm DC, Schnell JM, Lee C, Harrison MM | 2021 | Cell-type-specific chromatin occupancy by the pioneer factor Zelda drives key developmental transitions in *Drosophila* | http://www.ncbi.nlm.nih.gov/geo/query/acc.cgi?acc=GSE150931 | NCBI Gene Expression Omnibus, GSE150931 |

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
