## [Editor Report]

This is an important study that defines the role of the FruC transcription factor in key developmental decisions during neurogenesis in *Drosophila*. The authors combine genetics and genomic profiling to provide convincing evidence that FruC-regulated gene expression is correlated with changes in repressive histone marks. This study will be of wide general interest to the developmental biology field.

---

## [Decision Letter]

**Decision letter after peer review:**

Thank you for submitting your article "Low-level repressive histone marks fine-tune stemness gene transcription in neural stem cells" for consideration by *eLife*. Your article has been reviewed by 3 peer reviewers, and the evaluation has been overseen by Chris Doe, as Reviewing Editor and Marianne Bronner as the Senior Editor. The following individual involved in the review of your submission has agreed to reveal their identity: Irwin Davidson (Reviewer #2).

Essential revisions:

1) Add statistical analysis to Figures 5, 6, and S4. Alternatively, explain why this is not possible.

2) Provide a more direct measure of gene expression e.g. rt-qPCR, RNA-seq, or nascent RNA-seq. Alternatively, explain why this is not possible, and temper quantitative claims accordingly.

*Reviewer #1 (Recommendations for the authors):*

Comments:

1) The authors indicate that FruC is the predominant Fru isoform, but don't show the data for A and B, the other isoforms they examine. This data should be included to support their conclusion.

2) The authors use an Engrailed Repressor Domain (ERD) fusion with the FruC DNA binding domain to argue that FruC normally has a repressive function. The authors do not provide references or any support that the ERD is functional in type II neuroblasts and/or in their chimeric protein.

3) The authors argue in several places in their study that they validate antibodies by performing staining in the brat-null genotype. How does this validate the antibody? Is their independent data to support that is the expected expression pattern? Did the authors examine loss-of-function tissues?

4) The genotype of animals used in the study is often not clear. What is the Fru allele used other than the FruC allele? How were clones generated? Additionally, some of the shorthand is not clear in the figures. The FruSat allele they mention early on appears to be a large deletion. Could the loss of other genes be an issue in their study?

5) In the UMAP plots, the authors do not indicate what the colors show. For example, what is orange vs black in Figure 1F? Please check all figures.

6) Figure S1 B-The lower panel appears to be incorrect. The text said it shows the percentage of mitochondrial genes. This error also makes it difficult to determine the quality of the scRNA-seq data.

7) The microscopy images often point to small changes that are not obvious, but the authors use these to make conclusions. The authors should consider how to better show the regions to support their claims.

8) How were the data randomized in Figure 3A? How were enriched genes defined in D. Why are only the genes in parenthesis shown over the UMAP?

9) The way the authors indicate significant differences in some of the graphs is unclear. For example, in Figure 4E what does the ** indicate is different? The two lower rows of microscopy images do not appear different by eye. What are the arrows pointing to?

10) The authors should provide their code and data sets in a public repository. The authors should provide all the gene lists from the single cell and cut-and-run analyses. They should also have the statistical criteria used for their identification in a supplemental table.

11) How were the larvae genotyped? Was this using molecular techniques or phenotypic markers?

12) How were microscopy images quantified? Was the observer blind to genotype?

13) Have the authors considered finding genes with bivalent modification--those with both activating and repressive histone modifications. These genes are thought to be cell fate identity genes.

*Reviewer #2 (Recommendations for the authors):*

In the first section of the results, the authors use scRNA-seq to characterize the neural populations. As an alternative to the candidate approaches that the authors have used to identify key regulators can they also use unbiased computational regulome approaches (for example SCENIC) to predict other potential regulators that may be involved in key developmental decisions?

On page 9 and many other times throughout the text the authors state that Fru 'fine tunes' the expression of Notch effector genes. While the basis for this conclusion comes from the elegant genetic analyses, at no point do the authors directly measure gene expression in any of the genetic backgrounds. The authors should perform RT-qPCR analyses of the expression changes of key regulatory genes (or better still RNA-seq or even better nascent RNA-seq). It is important to do this to have some type of precise measure of transcriptional changes as the authors' model is based on subtle changes in H3K27me3. It would be important to quantify and correlate both processes more precisely.

A key aspect of the author's model is the changes in H3K27me3 levels at key target gene loci. However, the data illustrated in Figure 5 are rather confusing. First, the levels of H3K27ac are reduced upon Fru deletion as are the levels of H3K27me3. This issue is that the regions labelled by H3K27ac and H3K27me3 do not overlap. As regions with H3K27ac correspond to active enhancers it is clearly not these regions that are being targeted by H3K27me3. What is the evidence that the regions showing altered H3K27me3 are actually relevant for regulating target gene expression? Can the authors make a more global comparison of regions marked by K27ac and K27me3? In addition, the authors state that Fru promotes 'low levels' of K27me3 at its bound loci throughout the genome. How do the authors define 'low levels' low compared to what? What are the cut-off criteria that they use to define low versus high levels of HK27me3?

While the experiments in this paper are well carried out, the conclusions drawn should be carefully considered. The authors state 'Our data indicate that FruC likely functions together with PRC2 to dampen the expression of specific genes in mitotic neuroblasts by promoting low levels of H3K27me3 enrichment at their enhancers and promoters (Figure 7). We propose that local low-level enrichment of repressive histone marks can act to fine-tune gene expression.' As mentioned above there are no direct measurements of gene expression and in fact, what the data show is that changes in H3K27me3 correlate with altered gene expression. There is no direct molecular mechanism described. The above suggested that nascent RNA-seq would be useful to directly demonstrate that Fru directly affects transcription, but the authors could also perform Cut&Run for Pol II to ask which stages of transcription Pol II recruitment and PIC formation or pausing/elongation are targeted by Fru. For the moment, there is only a correlation with changes in histone modification. Either the authors should tone down the conclusions or perform additional experiments that do actually address a molecular mechanism by which Fru alters gene transcription. Either way, it is essential that changes in gene expression be directly assessed.

The authors should be careful with the figure annotation as lettering for panels is often missing.

*Reviewer #3 (Recommendations for the authors):*

The overall results of the work are compelling and experiments were performed thoroughly.

1) In the description of the results presented in Figure 1, it would be of relevance to clearly state the drivers used.

2) Figure 1 F – it would be easier to represent the expression of the different markers in the format of a dot plot so that colocalization of expression is easier to assess. If the authors choose to leave it as is a legend the colors should be added.

3) Figure 1 F – why these markers were used (TTFs) should be included as well as corresponding references.

4) Figure S1B -These plots could benefit from some more explanation.

5) In Figure 2 it would be of high relevance to include results for other fruitless isoforms, just as presented for fruc in Figure 2C.

6) In Figure 2 – The expression of markers is not easy to assess. For instance, in 2F only one Dpn+Ase- can be visualized and the Dpn staining is very dim and does not seem nuclear. If possible, include more representative images.

7) Quantifications should be presented for Figure 2D, E.

8) In Figure 2L: include values above the graph and not in the middle.

9) In Figure 2M-O it is barely possible to distinguish any Asense or Prospero staining. Possibly select better representative images, maybe include fewer arrows.

10) In this same section, where it is said "This result indicates that Fruc overexpression is sufficient to restore differentiation in brat-null brains" should read instead "is sufficient to partially restore (…)".

11) In figure 3G, what would be the expected random overlap, considering the big difference in peak numbers between the different datasets? (9301 peaks for Fru and 305 peaks for Su(H)).

12) In Figure 4, representative images of the quantifications would be relevant;

13) In FigS4 and Figure 5 it would be important to include the number of significant events in the volcano plots (which can be included in the figure itself);

14) One concern, and according to the Materials and methods, fruc::myc, which you used for the genome-wide studies, is a UAS/Gal4 system. Hence, this might mean that there is an overrepresentation of fruc binding sites that might be more subtle in biological situations. Did the authors perform any CUT&RUN experiments using the fruitless common antibody in a wild-type background?

15) Do the authors know what happens to PRC2-bound peaks in the absence of fruc? Or, in contrast, what happens to fruc-bound peaks when PRC2 subunits are absent?

---

## [Author Response]

Essential revisions:1) Add statistical analysis to Figures 5, 6, and S4. Alternatively, explain why this is not possible.

We thank the reviewer for the suggestion and apologize for not clearly labeling the Y-axis in these plots.

We have added the following statement in the figure caption "The horizontal line in the volcano plot represents -log10(0.05) = 1.301. All genes above this line has a FDR < 0.05".

2) Provide a more direct measure of gene expression e.g. rt-qPCR, RNA-seq, or nascent RNA-seq. Alternatively, explain why this is not possible, and temper quantitative claims accordingly.

We thank the reviewer for the suggestion.

We have added a panel of images in Figure 4F-N showing nascent transcripts of *Notch* and two of its downstream-effector genes *dpn* and *klu* as well as *CycE* as a negative control. We included quantification of nascent transcript foci for these genes in Figure 4O showing that knocking down *fru^C^* function significantly increases *Notch*, *dpn* and *klu* nascent transcript levels while not affecting the level of *CycE* nascent transcripts. We extended our analyses to demonstrate that knocking down the function of a key Notch downstream-effector gene *E(spl)mγ* in newborn neuroblast progeny suppressed increased supernumerary neuroblast formation in *numb^hypo^* brains heterozygous for *fru*.

We have included a paragraph in the text describing these new findings.

Reviewer #1 (Recommendations for the authors):Comments:1) The authors indicate that FruC is the predominant Fru isoform, but don't show the data for A and B, the other isoforms they examine. This data should be included to support their conclusion.

We thank the reviewer for the suggestion.

We have added images showing the expression of endogenously expressed Fru^A^::myc and Fru^B^::myc in Figure 2—figure supplement 1A-B.

2) The authors use an Engrailed Repressor Domain (ERD) fusion with the FruC DNA binding domain to argue that FruC normally has a repressive function. The authors do not provide references or any support that the ERD is functional in type II neuroblasts and/or in their chimeric protein.

We thank the reviewer for the comment.

We have added the following statements and references in the text to clarify the function of ERD in Fru^C,zf^::ERD:

"The ERD domain is well conserved in multiple classes of homeodomain proteins as well as many transcriptional repressors across the bilaterian divide and binds to the Groucho co-repressor protein to exert its repressor function (Smith and Jaynes 1996; Jiménez et al. 1997; Bürglin and Affolter 2016). Several previously published studies have used this strategy to demonstrate that neurogenetic transcription factors exert transcriptional repression function in neuroblasts (Xiao et al. 2012; Janssens et al. 2014; Bahrampour et al. 2017; Rives-Quinto et al. 2020)."

3) The authors argue in several places in their study that they validate antibodies by performing staining in the brat-null genotype. How does this validate the antibody? Is their independent data to support that is the expected expression pattern? Did the authors examine loss-of-function tissues?

We thank the reviewer for the comments.

In Figure 2B-C, we showed that Fru^C^::Myc and Fru^com^ co-localize with the neuroblast marker Dpn in wild-type larval brains. Most cells in *brat*-null brains are neuroblasts indicated by Dpn expression, and Fru^C^::Myc co-localizes with Dpn in *brat*-null brains confirming the specificity of the Myc antibody in detecting Fru^C^ expression in *brat*-null brains that carry the *fru^C^::Myc* allele.

Su(H) is the DNA-binding component of the Notch transcriptional activator complex and *dpn* is a conserved target of Notch signaling. Staining patterns of the Su(H) antibody and the Dpn antibody appear indistinguishable in *brat*-null brains validating the specificity of the Su(H) antibody in detecting active Notch signaling.

Staining patterns of the Trl antibody and the Dpn antibody appear indistinguishable in *brat*-null brains, which contain mostly neuroblasts. We concluded that Trl is expressed ubiquitously expressed in all neuroblasts, consistent with previously published studies reporting that Trl is constitutively expressed in all fly cell types examined to date.

All three antibodies have been previously validated to be specific for the protein they target in the study they were generated in.

4) The genotype of animals used in the study is often not clear. What is the Fru allele used other than the FruC allele? How were clones generated? Additionally, some of the shorthand is not clear in the figures. The FruSat allele they mention early on appears to be a large deletion. Could the loss of other genes be an issue in their study?

We thank the reviewer for the comments and apologize for the lack of clarity.

We have now included all genotypes for the experiments in the figure caption.

We showed that removing *fru* function using the *fru^sat15^* allele in the mosaic clones results in ectopic Dpn expression in newborn immature identical to knocking down *fru^C^* function by RNAi. Thus, the phenotype displayed by *fru^sat15^* homozygous neuroblasts should be specifically associated with loss of *fru^C^* function.

5) In the UMAP plots, the authors do not indicate what the colors show. For example, what is orange vs black in Figure 1F? Please check all figures.

We thank the reviewer for the comments and apologize for our mistakes.

We have added labels to the scale bar.

6) Figure S1 B-The lower panel appears to be incorrect. The text said it shows the percentage of mitochondrial genes. This error also makes it difficult to determine the quality of the scRNA-seq data.

We thank the reviewer for catching this and apologize for our mistakes.

We have now included the correct panel.

7) The microscopy images often point to small changes that are not obvious, but the authors use these to make conclusions. The authors should consider how to better show the regions to support their claims.

We thank the reviewer for the comments and apologize for our mistakes.

We have added additional information to the figure caption to describe the randomization process. Essentially, the same set of peaks in number and sizes is taken and shuffled to have new start coordinates in the genome.

Enriched genes in D are defined as described in the methods section:

“SCANPY’s rank_gene_groups was used with method=t-test-overestim_var to determine log-fold changes and adjusted p-values of expressed genes. Differentially expressed genes were defined as having |Fold Change| > 2 and adjusted p-value < 0.05, and invariant genes were defined as having |Fold Change| < 2.”

We have clarified that the genes in parenthesis are those that are bound by Su(H). The UMAPs display all genes bound, not just the ones shown in parentheses.

8) How were the data randomized in Figure 3A? How were enriched genes defined in D. Why are only the genes in parenthesis shown over the UMAP?

We thank the reviewer for their comments.

The ** indicated in the figure are described in the caption as indicating a p-value of <0.005 as determined by T-test as described in the methods section. This nomenclature is consistently used across all figures.

In the Figure 4B-D, white arrows point at representative type II neuroblasts marked by the Dpn^+^Ase^-^ marker combination. We added the description of white arrows in the caption. The panel of images in Figure 4C show more type II neuroblasts indicated by 3 white arrows than the panel of images in Figure 4D showing fewer lowest row in the panel the reviewer suggests is shown to have fewer type II neuroblasts type II neuroblasts indicated by 2 white arrows.

9) The way the authors indicate significant differences in some of the graphs is unclear. For example, in Figure 4E what does the ** indicate is different? The two lower rows of microscopy images do not appear different by eye. What are the arrows pointing to?

We thank the reviewer for their comments.

We have uploaded all code used for data analyses onto github. Data sets including processed data are uploaded onto the GEO site. We have also added Supplemental Table 1 with all raw data and figure statistics.

10) The authors should provide their code and data sets in a public repository. The authors should provide all the gene lists from the single cell and cut-and-run analyses. They should also have the statistical criteria used for their identification in a supplemental table.

We thank the reviewer for this comment.

Homozygous mutant larvae were identified by lack of Act-GFP expression or lack of the Tubby phenotype which is indicative of the balancer chromosome.

11) How were the larvae genotyped? Was this using molecular techniques or phenotypic markers?

We thank the reviewer for this comment.

We have added additional description to the methods section to describe how images were quantified, including new smFISH data.

The observers were blind to the genotypes.

12) How were microscopy images quantified? Was the observer blind to genotype?

We thank the reviewer for the suggestion.

Identification and characterization of these putative bivalent genes will be described in a future study.

13) Have the authors considered finding genes with bivalent modification--those with both activating and repressive histone modifications. These genes are thought to be cell fate identity genes.

We thank the reviewer for the comment.

While the suggestion to identify key regulators of neuronal stem population is great, it is outside the scope of the current study. We hope the scRNA-seq data we generated will be useful to other in doing further studies on factors that regulate neural stem cell function.

Reviewer #2 (Recommendations for the authors):In the first section of the results, the authors use scRNA-seq to characterize the neural populations. As an alternative to the candidate approaches that the authors have used to identify key regulators can they also use unbiased computational regulome approaches (for example SCENIC) to predict other potential regulators that may be involved in key developmental decisions?On page 9 and many other times throughout the text the authors state that Fru 'fine tunes' the expression of Notch effector genes. While the basis for this conclusion comes from the elegant genetic analyses, at no point do the authors directly measure gene expression in any of the genetic backgrounds. The authors should perform RT-qPCR analyses of the expression changes of key regulatory genes (or better still RNA-seq or even better nascent RNA-seq). It is important to do this to have some type of precise measure of transcriptional changes as the authors' model is based on subtle changes in H3K27me3. It would be important to quantify and correlate both processes more precisely.

We thank the reviewer for the suggestion and agree with the reviewer that a direct measure of gene transcription will greatly strengthen our claims.

We have added a panel of images in Figure 4F-N showing nascent transcripts of *Notch* and two of its downstream-effector genes *dpn* and *klu* as well as *CycE* as a negative control. We included quantification of nascent transcript foci for these genes in Figure 4O showing that knocking down *fru^C^* function significantly increases *Notch*, *dpn* and *klu* nascent transcript levels while not affecting the level of *CycE* nascent transcripts. We extended our analyses to demonstrate that knocking down the function of a key Notch downstream-effector gene *E(spl)mγ* in newborn neuroblast progeny suppressed increased supernumerary neuroblast formation in *numb^hypo^* brains heterozygous for *fru*.

We have included a paragraph in the text describing these new findings.

A key aspect of the author's model is the changes in H3K27me3 levels at key target gene loci. However, the data illustrated in Figure 5 are rather confusing. First, the levels of H3K27ac are reduced upon Fru deletion as are the levels of H3K27me3. This issue is that the regions labelled by H3K27ac and H3K27me3 do not overlap. As regions with H3K27ac correspond to active enhancers it is clearly not these regions that are being targeted by H3K27me3. What is the evidence that the regions showing altered H3K27me3 are actually relevant for regulating target gene expression? Can the authors make a more global comparison of regions marked by K27ac and K27me3? In addition, the authors state that Fru promotes 'low levels' of K27me3 at its bound loci throughout the genome. How do the authors define 'low levels' low compared to what? What are the cut-off criteria that they use to define low versus high levels of HK27me3?

We thank the reviewer for the comment.

Three pieces of evidence support our model that Fru^C^ functions through low levels of H3K27me3 to finely tune target gene transcription. First, we showed in Figure 5A-F that changes in H3K27me3 levels at many Fru^C^-bound loci directly correlate with *fru^C^* function. Second, we showed in Figure 6A-C that PRC2 components are enriched at thousands of Fru^C^-bound peaks. Third, we showed in Figure 6D-G that reducing PRC2 function increases Notch signaling during asymmetric neuroblast division phenocopying the effect of loss of *fru^C^* function.

We also added a plot to Figure 5G showing low versus high levels of H3K27me3. To better define what low levels of H3K27me3 are. Additionally, we have added Figure 5B, showing how the traditional Polycomb regions which show high levels of H3K27me3 are largely unaffected.

While the experiments in this paper are well carried out, the conclusions drawn should be carefully considered. The authors state 'Our data indicate that FruC likely functions together with PRC2 to dampen the expression of specific genes in mitotic neuroblasts by promoting low levels of H3K27me3 enrichment at their enhancers and promoters (Figure 7). We propose that local low-level enrichment of repressive histone marks can act to fine-tune gene expression.' As mentioned above there are no direct measurements of gene expression and in fact, what the data show is that changes in H3K27me3 correlate with altered gene expression. There is no direct molecular mechanism described. The above suggested that nascent RNA-seq would be useful to directly demonstrate that Fru directly affects transcription, but the authors could also perform Cut&Run for Pol II to ask which stages of transcription Pol II recruitment and PIC formation or pausing/elongation are targeted by Fru. For the moment, there is only a correlation with changes in histone modification. Either the authors should tone down the conclusions or perform additional experiments that do actually address a molecular mechanism by which Fru alters gene transcription. Either way, it is essential that changes in gene expression be directly assessed.

We thank the reviewer for the suggestion.

We have added a panel of images in Figure 4F-N showing nascent transcripts of *Notch* and two of its downstream-effector genes *dpn* and *klu* as well as *CycE* as a negative control. We included quantification of nascent transcript foci for these genes in Figure 4O showing that knocking down *fru^C^* function significantly increases *Notch*, *dpn* and *klu* nascent transcript levels while not affecting the level of *CycE* nascent transcripts. We extended our analyses to demonstrate that knocking down the function of a key Notch downstream-effector gene *E(spl)mγ* in newborn neuroblast progeny suppressed increased supernumerary neuroblast formation in *numb^hypo^* brains heterozygous for *fru*.

We have included a paragraph in the text describing these new findings.

The authors should be careful with the figure annotation as lettering for panels is often missing.

We thank the reviewer for the suggestion and apologize for our mistakes.

We have corrected all errors.

Reviewer #3 (Recommendations for the authors):The overall results of the work are compelling and experiments were performed thoroughly.1) In the description of the results presented in Figure 1, it would be of relevance to clearly state the drivers used.

We thank the reviewer for the comments and apologize for our mistakes.

We added the genotype of the driver used in Figure 1 in the figure caption.

2) Figure 1 F – it would be easier to represent the expression of the different markers in the format of a dot plot so that colocalization of expression is easier to assess. If the authors choose to leave it as is a legend the colors should be added.

We thank the reviewer for the comments and apologize for our mistakes.

We added the scale bar in the figure.

3) Figure 1 F – why these markers were used (TTFs) should be included as well as corresponding references.

We thank the reviewer for the comments and apologize for our mistakes.

We added the corresponding references for temporal transcription factors used in this analysis.

4) Figure S1B -These plots could benefit from some more explanation.

We thank the reviewer for the comments and apologize for our mistakes.

We have updated the plot for Figure 1—figure supplement 1B and provided explanation in the figure caption.

5) In Figure 2 it would be of high relevance to include results for other fruitless isoforms, just as presented for fruc in Figure 2C.

We thank the reviewer for the suggestion and apologize for our mistakes.

We have added images showing the expression of endogenously expressed Fru^A^::myc and Fru^B^::myc in Figure 2—figure supplement 1A-B.

6) In Figure 2 – The expression of markers is not easy to assess. For instance, in 2F only one Dpn+Ase- can be visualized and the Dpn staining is very dim and does not seem nuclear. If possible, include more representative images.

We thank the reviewer for the comment.

The images in Figure 1F-G reflect the multilayered control of Dpn expression in neuroblast progeny by translational and post-translational regulatory mechanisms (Komori et al. *G&D* 2018). Consequently, the majority of newborn neuroblast progeny (85.7%) show undetectable Dpn expression shown in Figure 1G. By contrast, the majority of newborn neuroblast progeny (94.4%) show low levels of Dpn expression following *fru^C^* knockdown shown in Figure 1F.

7) Quantifications should be presented for Figure 2D, E.

We thank the reviewer for the suggestion and apologize for our mistakes.

1-2 immature INPs in 100% of *fru^sat15^* clones showed low Dpn expression whereas no immature INPs showed detectable Dpn expression.

8) In Figure 2L: include values above the graph and not in the middle.

We thank the reviewer for the suggestion.

We added the value above the plot. in Figure 2L.

9) In Figure 2M-O it is barely possible to distinguish any Asense or Prospero staining. Possibly select better representative images, maybe include fewer arrows.

We thank the reviewer for the suggestion.

The progeny of type II neuroblasts in *brat*-null brains revert to become supernumerary type II neuroblasts instead of committing to an INP identity and generating GMCs. For this reason, there is no detectable Ase and Pros expression in the dorsal region of *brat*-null brains shown in Figure 2M.

10) In this same section, where it is said "This result indicates that Fruc overexpression is sufficient to restore differentiation in brat-null brains" should read instead "is sufficient to partially restore (…)".

We thank the reviewer for this suggestion.

We toned down the description in the text and the figure caption.

11) In figure 3G, what would be the expected random overlap, considering the big difference in peak numbers between the different datasets? (9301 peaks for Fru and 305 peaks for Su(H)).

We thank the reviewer for this suggestion.

We have added an additional random randomized peak set to demonstrate that Su(H) binds more neuroblast enriched genes than would be expected based on 305 randomly bound peaks but binds similar numbers as would be expected for invariant or immature INP enriched genes.

12) In Figure 4, representative images of the quantifications would be relevant;

We thank the reviewer for this suggestion.

We added all images shown in the quantification in Figure 4—figure supplement 1A-E, H-L.

13) In FigS4 and Figure 5 it would be important to include the number of significant events in the volcano plots (which can be included in the figure itself);

We thank the reviewer for this suggestion.

We added the number of events in the volcano plots.

14) One concern, and according to the Materials and methods, fruc::myc, which you used for the genome-wide studies, is a UAS/Gal4 system. Hence, this might mean that there is an overrepresentation of fruc binding sites that might be more subtle in biological situations. Did the authors perform any CUT&RUN experiments using the fruitless common antibody in a wild-type background?

We thank the reviewer for this suggestion and apologize for the confusion.

We clarified in the text that *fru^C^::Myc* is a knock-in allele of *fru^C^*.